# Recent Advances on Starch-Based Adsorbents for Heavy Metal and Emerging Pollutant Remediation

**DOI:** 10.3390/polym17010015

**Published:** 2024-12-25

**Authors:** Talles B. Costa, Pedro M. C. Matias, Mohit Sharma, Dina Murtinho, Derval S. Rosa, Artur J. M. Valente

**Affiliations:** 1CQC-IMS, Department of Chemistry, University of Coimbra, 3004-535 Coimbra, Portugal; tallesbarcelos@hotmail.com (T.B.C.); petermatias1998@gmail.com (P.M.C.M.); dmurtinho@ci.uc.pt (D.M.); 2Engineering, Modeling, and Applied Social Sciences Center (CECS), Federal University of ABC, Santo André 09280-560, SP, Brazil; derval.rosa@ufabc.edu.br; 3CERES, Department of Chemical Engineering, University of Coimbra, 3030-790 Coimbra, Portugal; mohit@eq.uc.pt

**Keywords:** starch, starch-based materials, adsorption, heavy metals, emerging pollutants, water remediation

## Abstract

Starch is one of the most abundant polysaccharides in nature and has a high potential for application in several fields, including effluent treatment as an adsorbent. Starch has a unique structure, with zones of different crystallinity and a glycosidic structure containing hydroxyl groups. This configuration allows a wide range of interactions with pollutants of different degrees of hydrophilicity, which includes from hydrogen bonding to hydrophobic interactions. This review article aims to survey the use of starch in the synthesis of diverse adsorbents, in forms from nanoparticles to blends, and evaluates their performance in terms of amount of pollutant adsorbed and removal efficiency. A critical analysis of the materials developed, and the results obtained is also presented. Finally, the review provides an outlook on how this polysaccharide can be used more effectively and efficiently in remediation efforts in the near future.

## 1. Introduction

The scarcity of natural resources, coupled with the growing limitations faced by countries in accessing raw materials, is ultimately jeopardizing their sovereignty, and presenting escalating challenges to society [1]. Among these challenges, water pollution stands out as a critical issue, highlighting the need to preserve this vital resource. Additionally, the presence of various pollutants in the water, as a result of anthropogenic activities, demands urgent attention and effective solutions.

Water pollution is a serious global environmental problem that harms ecosystems, biodiversity, and human health. It can cause numerous diseases and, in cases of prolonged exposure, lead to death. Water pollution therefore threatens the safe supply of drinking water to populations, which can affect up to 5.5 billion people worldwide by 2100 [2]. It is therefore imperative to promote water decontamination, as many pollutants are toxic, carcinogenic, and teratogenic, even in low concentrations. These contaminations include organic molecules (e.g., dyes, pesticides, pharmaceutical and personal care products, biotoxins, food additives, polycyclic aromatic hydrocarbons, volatile organic compounds, disinfection byproducts or perfluorinated compounds) and inorganic substances (i.e., heavy metals, non-metals and radionuclides) [3,4,5,6]. Many of these compounds are labeled emerging pollutants [7,8,9,10,11,12].

Today’s reality calls for the development of methodologies that address the problem in two different ways: developing efficient methodologies for pollutant removal from water to clean up water and protect aquatic ecosystems, and ensuring the sustained reuse of those compounds.

Various techniques are used to remove pollutants, and it is increasingly common for removal to be followed by reuse. However, the most effective pollutant removal techniques, including advanced oxidation processes, and hybrid systems such as electrically enhanced membrane bioreactors [13,14], are also the least selective. While some technologies are more suitable for targeting specific contaminants than others [15], most of them have high costs, low efficiency/selectivity, and produce large amounts of waste. As an alternative, adsorption technologies are more advantageous and sustainable due to their durability, high removal efficiency, versatility, low cost, and easy operation. In fact, adsorption methodologies, especially those based on multifunctional adsorbents, offer the best balance between removal efficiency and reuse.

Among the polymers that have attracted greater attention in the literature are polysaccharides, a relevant class of biological and biodegradable polymers obtained from renewable sources (i.e., plant, animal, microbe or algae) [16]. These include starch, gums (e.g., gum Arabic), cellulose (the most abundant polysaccharide on Earth, based on D-glucose units linked by β(1→4) glycosidic bonds [17]), chitosan (the second most present polysaccharide in nature, being a linear cationic polymer consisting of D-glucosamine moieties and a few *N*-acetyl-D-glucosamine units [18]) and alginate (a linear polymer formed by β-D-mannuronic and α-L-guluronic acid monomers [19]). Due to their nontoxicity, low cost, natural availability, biodegradability, biocompatibility, and renewable nature, these biopolymers are considered promising alternatives to traditional synthetic polymers for many applications, including adsorption, catalysis, and biomedical purposes [4]. This review, in particular, focuses on starch, a polysaccharide that is quite abundant in nature as a primary storage polymer in plants [20,21], and has the capacity to be used as an adsorbent either on its own (with or without functionalization) [22], as a blend or composite [23], or even in nanoparticles [24,25], and whose potential for application as an adsorbent is, in our view, still underestimated.

Therefore, this paper reviews the recent progress (since 2023) in materials containing native or modified starch and other functional materials (used to overcome starch limitations), focusing on their synthesis, properties, and performance for application in the adsorption removal of heavy metals and emerging organic contaminants from water. Furthermore, the types of interactions that control the removal of pollutants, as well as the equilibrium, kinetics, and thermodynamics of the adsorption processes involving various starch-based adsorbents, will also be summarized.

## 2. Starch

Starch is a natural, biodegradable, renewable, and highly abundant polysaccharide. Its industry has steadily grown over the years, with starch production in the European Union (EU) increasing from 8.7 million tons in 2004 to 9.2 million tons in 2023. Producing this volume requires processing approximately 22 million tons of agricultural raw materials, comprising 45% wheat, 39% maize, and 16% potatoes and other crops. In 2023, starch consumption in the EU reached 7.2 million tons, distributed as 48% starch sweeteners, 34% native starch, and 18% modified starch. The primary applications of starch were in the corrugating and paper industries (33%), food and non-food use (61%), and the chemicals sector (6%) [26].

### 2.1. Structure of Starch

Starch is mainly composed of two polysaccharides, amylose and amylopectin, both of which are glucose polymers with distinct structural characteristics that define the unique properties of starch. Amylose is a linear polymer consisting of α-D-glucopyranose units linked by 1,4 glycosidic bonds. This linear structure allows amylose to adopt helical conformations. In contrast, amylopectin is highly branched, containing both 1,4 and 1,6 glycosidic linkages. This branching creates bulkier regions within starch granules. Amylopectin is typically the predominant component of starch, comprising up to 95% in certain types [27].

The size, shape, structure, amylose-to-amylopectin ratio, and degree of crystallinity of starch granules are influenced by their botanical origin. The degree of crystallinity, determined by the amylose-to-amylopectin ratio and starch origin, plays a critical role in influencing the functionality and processing behavior of starch [28,29]. At the supramolecular level, amylose and amylopectin form granules that can vary widely in size, reaching diameters of up to 100 μm. These granules grow outward from a central nucleus known as the hilum. During granule development, amylose integrates into the structure, with branching being more prominent in less crystalline, semicrystalline, or amorphous regions. In contrast, more crystalline regions are characterized by adjacent chains forming double helices of the linear or branched polymer, which are packed tightly in ordered arrays (Figure 1).

### 2.2. Properties of Starch

The structure of starch, especially the amylose-to-amylopectin ratio and the supramolecular organization of its granules, is fundamental to its physical and chemical properties. Key properties of starch include the following:

*Crystallinity*.

The crystalline regions of starch contribute to its rigidity and insolubility at room temperature (r.t.). However, these regions can be altered through various treatments, including thermal processing, chemical modifications, or a combination of both. Such modifications are used to tailor the properties of starch for specific applications [31].

*Gelatinization*.

One of the most important properties of starch is its ability to gelatinize when heated in the presence of water. During gelatinization, starch granules absorb water, swell, and eventually rupture, resulting in the loss of their crystalline structure. This process occurs as the crystalline regions of starch are disrupted by increased hydrogen bonding between water molecules and the hydroxyl groups of starch, leading to granular swelling. Gelatinization is primarily driven by amylopectin, as its branched structure makes it more susceptible to disruption under heat and moisture. Simultaneously, amylose molecules diffuse out of the swollen granules, contributing to an increase in the viscosity of the solution. Variations in the swelling and pasting properties of starch are largely attributed to differences in the length distribution of the amylopectin unit chain [32].

*Retrogradation*.

Retrogradation refers to the process by which gelatinized starch molecules transition from a disordered to an ordered state. As the starch cools, the decrease in kinetic energy allows the molecules to reassociate and form a network. Amylose, with its linear structure, readily realigns to create ordered structures and plays a critical role in retrogradation. In contrast, the branched structure of amylopectin limits its mobility after gelatinization, requiring more time to reorganize into an ordered state. Consequently, starches with higher amylose content tend to retrograde faster and more extensively than those with higher amylopectin content, influencing the texture and shelf life of starch-based products [33].

Over extended storage, reversible recrystallization of amylopectin occurs, increasing the rigidity of swollen granules embedded within the continuous amylose network. The linear structure of amylose also helps in forming films and elastic gels that resist disintegration, making it particularly valuable for applications such as biodegradable packaging materials [29,34]. On the other hand, waxy starch, which is predominantly amylopectin, produces gels with greater stickiness, adhesiveness, and penetrability [35].

### 2.3. Limitations of the Use of Native Starch

Despite its high availability and unique properties, starch has several limitations in its natural form. For instance, it is insoluble in most solvents at room temperature and is sensitive to changes in pH, temperature, and shear forces. These characteristics can limit its effectiveness in certain industrial applications. For example, retrogradation can affect the stability of starch-based adhesives or coatings, reducing their flexibility and durability over time [33].

Native starch also has relatively poor mechanical properties, particularly in strength and flexibility, due to its semicrystalline structure. This restricts its use in applications requiring high mechanical performance, such as bioplastics or structural materials. In industries, such as papermaking or packaging, where starch is used as a surface sizing agent or coating additive, retrogradation at lower temperatures can hinder uniform application and result in brittle products upon cooling. Moreover, starch’s hygroscopic nature can lead to moisture absorption, causing swelling, loss of strength, or stickiness in starch-based products. Likewise, in food applications, retrogradation must be carefully controlled to meet specific texture requirements. For example, foods like spaghetti benefit from moderate retrogradation, while others, such as noodles, require its suppression to achieve the desired texture [36].

To address these challenges, the modification of the crystalline structure of starch using amylase enzymes can reduce amylose chain length, thereby lowering retrogradation [37,38]. Additionally, chemical modifications that introduce functional groups, such as amino, carbonyl, carboxyl, ester, or cationic groups, enhance starch’s properties. These modified starches find applications in wastewater treatment, medicine, papermaking (as strength additives), and the textile and petrochemical industries. They also improve the adsorption affinity of starch for heavy metal ions and other pollutants [31,39,40,41,42,43,44,45].

## 3. Starch-Based Materials: Synthesis and Properties

The separation of components by adsorption for environmental remediation is important in the context of climate emergency and has been widely applied at the industrial level. Due to its advantages, many biopolymer-based adsorbents have been investigated, with starch emerging as a promising option due to its natural abundance (it is one of the three most abundant polysaccharides and the most common digestible storage carbohydrate polymer in plants), cost-effectiveness, reusability, biodegradability, and richness in hydroxyl groups [46,47,48,49]. Alternatively, other functional chemically modified starches, such as carboxymethyl starch (CMS) [50,51,52,53], carboxyl or dialdehyde-containing oxidized starches [54,55,56,57], starch xanthate [58], starch ester [59,60], hydroxypropyl starch [61] and hydroxypropyl sulfonated starch [62], can also be used to overcome the limited variety of functional groups of native starch. Additionally, combining starch/modified starch with other functional materials to produce starch-based composites and blends is a common strategy applied today to achieve economical multifunctional high molecular weight insoluble materials, endowed with advantageous physicochemical properties, i.e., high specific surface area and porosity. These properties promote mass transport and contact between active sites and external species (such as, heavy metals, pharmaceuticals, dyes, pesticides, endocrine disrupting compounds and microplastics) more efficiently than native starch, which has poor morphological and surface properties as well as adsorption capacities [48].

### 3.1. Synthesis Methods

To satisfy current concerns, in recent years, numerous materials have been synthesized using starch as a raw material through various strategies. The different synthetic approaches not only tend to meet Green and Sustainable Chemistry criteria, but also to formulate materials with more processable morphologies than simple powders (e.g., hydrogels [63,64,65,66,67,68,69], aerogels [53,70,71], cryogels [71], membranes [72] and films [73,74,75]), which is critical for their large-scale application.

The modification of the hydroxyl groups of starch involves overcoming physicochemical forces to make them accessible, since they are located within a layered microstructure [48]. With this aim, in recent years, starch-based materials with high performance for adsorption have been prepared through various methods (sonochemistry, electromagnetic radiation, hydrothermal and solvothermal conventional heating, coprecipitation, dry-heating) and diverse types of modifications (through different chemical reactions, grafting, chemical/physical crosslinking, carbonization/pyrolysis, internal and external gelation). The synthesis strategies are described below, and Table 1, Table 2 and Table 3 (Section 4) summarize the synthetic methods for starch-based materials reported since the beginning of 2023 for the removal of metals, dyes and other emerging pollutants, respectively.

#### 3.1.1. Hydrothermal and Solvothermal Conventional Synthesis

This procedure can require conditions ranging from mild to severe, as it involves the conventional heating of a reaction mixture (including starch/modified starch and other components, such as a crosslinker, catalysts, modification agents or matrix-forming compounds) in an organic solvent (solvothermal) or water/aqueous solutions (hydrothermal) on a hotplate or in an oven at different temperatures (from room temperature to 200 °C [76,77,78,79,80,81]) for a certain period of time and under high pressure [71,81], atmospheric pressure or in an inert atmosphere in a closed container [54,56,82,83,84,85]. In the last two years, conventional heating has been the most widely adopted physical methodology for the preparation of starch-based materials. Although different solvents (e.g., ethanol [54,78,82,85,86], methanol [87], tetrahydrofuran [83], isopropanol [56], *N*,*N*-dimethylformamide [88,89], acetone [90] and glycerol [81,91]) can be used to disperse or dissolve components, water/aqueous acid or basic solutions and the gelatinization of starch by heating in an aqueous medium are preferred [53,54,82,88,92,93].

#### 3.1.2. Sonochemistry

The application of high-energy ultrasonic waves (of frequency between 20 kHz and 10 MHz) to particles much smaller than their wavelength promotes continuous rarefaction and compression in a sample. During rarefaction, the pressure becomes lower than the vapor pressure of the solvents, promoting the formation, growth, and consequent collapse of bubbles, as they become unstable with growth. This phenomenon is called acoustic cavitation and produces rapid release and gain of energy/heat, capable of accelerating a physicochemical process [94,95]. The pressure and temperature gradients during the rapid collapse of the bubbles generate shear forces that can break particles and chemical bonds, such as the α-1,4 and α-1,6 glycosidic bonds in starch, reducing its size and molar mass, and altering its structure and rheological properties [96]. This pretreatment procedure can also work as an activation method, increasing the adsorption capacity of starch-based materials, such as that of crosslinked cationic potato starch granules for ibuprofen, from 345 to 493 mg g^−1^ [97]. Recently, Li et al. [98] and Hakke et al. [99] also adopted an ultrasound-assisted strategy to promote acid or enzymatic hydrolysis of starch.

Ultrasound irradiation has also been used as a powerful, green and affordable technique to facilitate the dissolution of compounds (such as starch in water [100] or MnSO_4_ and Na_3_C_6_H_5_O_7_ in water for the preparation of magnetic starch-based composite containing Fe-Mn nanoclusters [101]); to prevent particle agglomeration and promote good dispersion of solid particles in a solvent (such as in the synthesis of acid hydrolyzed starch [102], starch/itaconic acid/acrylic acid-based hydrogel containing modified cellulosic nanofiber-ZnO [85], and starch/montmorillonite-MnFe_2_O_4_-ZIF-67 [76]); to clean and degas procedures [103]; and to foster reactions, e.g., the surface reaction between *N*-vinylformamide-starch graft copolymer and Fe_3_Mn_3_O_8_ magnetic nanoparticles [104] or the free radical polymerization to prepare starch-containing polyacrylate hydrogels [105,106].

#### 3.1.3. Electromagnetic Radiation Induced Synthesis

Electromagnetic waves with different energies (i.e., microwave < ultraviolet (UV) < γ-rays by energy order) can promote the formation of starch-based materials. Microwave-assisted synthesis can speed up the progress of a reaction compared to conventional heating, due to the microwave dielectric heating (at a frequency of 2.45 GHz) by dipolar polarization and ionic conduction mechanisms that allow for the in loco, direct, rapid and uniform generation of heat (unlike conventional and ultrasonic heating [107]), without requiring a direct contact between the microwave energy source and the sample [108,109]. Kangmennaa et al. [110] used this approach to promote graft polymerization between (V_2_O_5_)-starch and acrylonitrile to obtain V_2_O_5_-St-g-PAN as methylene blue adsorbent.

UV light [70,111,112] and γ-rays [91,113,114] can be used as an alternative to conventional heating for the induction of free radical reactions [115]. They are very efficient, fast, and environmentally friendly techniques that take place at room temperature to avoid thermal degradation [112,113]. Photoinitiators (such as 2,2-dimethoxy-2-phenylacetophenone [111] or 2,2-diethoxyacetophenone [70]) upon absorption of UV radiation of the appropriate wavelength (typically 300–400 nm) dissociate and promote polymerization reactions. On the other hand, ionizing γ-radiation works as an initiator for polymerization, crosslinking and grafting reactions in the preparation of starch-based materials for high performance applications. It promotes the breaking of covalent bonds to generate free radicals, which, consequently, can lead to the occurrence of new chemical reactions between molecules. This technique avoids the use of chemical initiators, facilitating the purification steps. In the synthesis of starch-acrylic acid-nanohalloysite [114] and starch/poly[2-(dimethylamino) ethyl methacrylate]/bismuth (St/DMAEMA/Bi) [91] composites, a cobalt-60 cell was used as γ-radiation source, supplying a radiation dose of 24 and 30 kGray, respectively, to form macroradicals from starch and ^●^OH radicals from water to promote free radical copolymerization. As an equally advantageous alternative to UV and γ-ray polymerization, electron beam irradiation can also generate radical formation and increase the crosslinking degree [74].

#### 3.1.4. Coprecipitation Method

The coprecipitation method consists of adding a precipitating agent to a solution containing at least two types of cations to produce the precipitation of a material after the reaction. In this way, normally soluble compounds can be adsorbed onto the nuclei or crystals under precipitation conditions and integrate into the solid structure as a single phase [116,117]. This low-cost strategy allows the preparation of a solid with a uniform chemical composition and a small, uniform particle size (nanoparticles) [118,119,120]. Manzoor et al. [64] and Niroumand et al. [76] synthesized starch-based nanomaterials by this method. In the first case, a mixture of ZnSO_4_ and CuSO_4_ was added dropwise to an aqueous solution of Na_2_CO_3_ precipitating agent at 60 °C and pH = 11, allowing metal precipitation, which, after calcination gave rise to a copper-doped zinc oxide nanomaterial, which was subsequently used for the preparation of poly(vinyl alcohol)/starch/copper-doped zinc oxide hydrogel (PVA/starch/NC/HG) [64]. In the second situation, montmorillonite (MMT) and starch (St) were added to an aqueous solution containing Fe(III) and Mn(II) ions. After stirring at 85 °C and pH = 10–11 (adjusted by adding NaOH), the St/MMT-MnFe_2_O_4_ nanocomposite adsorbent was formed after 3 h and collected using a magnet [76].

#### 3.1.5. Dry-Heating Method

Dry-heating treatment involves heating a dried mixture at (110–150) °C. It is a safe and operationally easy physical modification strategy, requiring temperature, time, and pressure optimization. It does not destroy the starch granular structure but modifies its physicochemical properties [121,122]. Recently, Liu et al. [79] prepared and dried citrate esterified starches (ES-St) and then subjected the fully pulverized mixture to dry-heating treatment at 130 °C for 6 h. Afterwards, a starch-based hydrogel was synthesized by grafting and crosslinking using the obtained ES-St, microcrystalline cellulose, and acrylic acid (AA), and applied for the efficient removal of Cu(II) and methylene blue. Similarly, Kim et al. [78] dry heated modified ethanol-treated corn starch with phytic acid and/or citric acid and non-modified starches at 130 °C for 12 h. Among the prepared materials, starch treated with both acids showed the best adsorption capacity for Cu(II) removal (38 vs. 0.11, 0.49, 2, and 36 mg g^−1^ for the native, ethanol-treated, ethanol-phytic acid treated, and ethanol-citric acid treated starches, respectively).

#### 3.1.6. Internal and External Gelation

This methodology is widely used for the preparation of alginate-based emulsion gels [123,124]. However, starch can be included in the structure. Starch/alginate hydrogels were synthesized by Lencina et al. [71], using both internal and external gelation mechanisms, based on the ion exchange reaction between Na(I) in the alginate structure and Ca(II). The internal gelation involved mixing both biopolymers (3% *w*/*w* biopolymer content, 25% *w*/*w* of which is alginate) and the crosslinking agent (Ca(II) salt), and then placing the mixture into molds. Subsequently, H_2_CO_3_ production in a CO_2_ atmosphere led to a decrease in pH, promoting the gradual release of Ca(II) ions from the salt and the internal crosslinking of alginate through a kinetically controlled process, not dependent on diffusive effects. On the other hand, external gelation is purely diffusive, and consists of adding the biopolymer mixture directly to the Ca(II) crosslinking solution. As Ca(II) is totally available, the hydrogel formation occurs very quickly on the external surface, while on the inside the gelation is a time-dependent radial process. For this reason, beads are left submerged in the crosslinker solution for a longer period, to ensure complete gelation. Lencina et al. selected external gelation as the best option, as it allowed simpler and more economical material preparation. Despite the synthesis strategy, the drying method can also influence the final properties of the materials. Thus, if the solvent molecules in the hydrogel pores are replaced by a gaseous medium (e.g., through supercritical CO_2_ drying) an aerogel is obtained [125]; while cryogels can be obtained by freeze-drying, where freezing excludes the solvent molecules, forming a gel in between the ice crystals [126]. In the cited work [71], aerogel beads presented the highest adsorption capacity and efficiency for Cu(II) (40 mg g^−1^ and 92% vs. 20 mg g^−1^ and 46% for cryogel beads), due to their high surface area and because supercritical CO_2_ better preserves the microstructure.

#### 3.1.7. Graft Polymerization

Graft polymerization is the most common and efficient chemical modification strategy used to obtain starch-based materials with superior physicochemical properties and high adsorptive performance, since it allows the addition of new functionalities (such as, amino, acetyl, carboxyl, amide, hydroxypropyl) by modifying the hydroxyl groups of starch [48]. It consists of the chemical bonding and polymerization of monomers in a backbone polymer chain, forming secondary chains, and, consequently, a graft copolymer that has mixed properties of at least two different polymers [127]. Grafting can occur by two different mechanisms: free radical and ionic [128]. For starch-based material preparation, free radical grafting is the most widely used [58,83,84,104,129,130,131,132,133,134] and is triggered by initiators such as potassium persulfate (KPS) [58,84,104,115,133], ammonium persulfate [66,68,131,132], azobisisobutyronitrile (AIBN) [83], ascorbic acid/H_2_O_2_ [52], potassium peroxymonosulfate [134], 1,1′-azobis-(cyclohexane carbonitrile) [135], ceric ammonium nitrate [67] or the Fenton solution (H_2_O_2_/Fe(II)) [136].

#### 3.1.8. Crosslinking and Chemical Modification Reactions

Chemical crosslinking, like grafting, allows new functional groups to be introduced in the material through a chemical modification process. For this, a multifunctional compound (crosslinker) is used to bind two different moieties [107], such as citric acid or trisodium citrate salt [78,93,133,137,138,139,140], *N*,*N*’-methylenebisacrylamide (MBA) [51,54,58,70,79,84,85,105,132], 1,6-hexanediamine [53], glycidyl methacrylate [83], boric acid [64], glutaraldehyde [141], sodium tri-metaphosphate [52], sodium tripolyphosphate [142], sodium dodecyl sulfate (SDS) [131], epichlorohydrin (EPI) [59,142,143,144], 2-(dimethylamino)ethyl methacrylate) [91], 1,4-butanediol diglycidylether (BDE) [145], triphenylmethane-4,4′,4″-triisocyanate (TMT) [89], CaCO_3_ and CaCl_2_ [71]. Despite the reactive groups, crosslinkers can also contain other functional groups that will be incorporated into the final structure. The reaction product will be a functionally improved crosslinked starch-based material, with greater physicochemical stability, since the new covalent bonds formed during chemical crosslinking together with the hydrogen bonds occurring in the starch-based particles will strengthen the final polymer network [48,146,147]. The association of starch or modified starch derivatives with other functional materials in the absence of any external agent and involving the establishment not only of hydrogen bonds but also of other weak non-covalent bonds, such as metal coordination, ionic and hydrophobic interactions, is known as physical crosslinking [144,148,149,150,151].

To promote the functionalization of starch-based materials or even chemical crosslinking processes, different chemical reactions involving the hydroxyl groups in starch can be carried out, such as esterification (with succinic acid [90], itaconic acid [85,132,133], citric acid/trisodium citrate [78,79,93,133,137,138,139,140,152] and octenylsuccinic anhydride [70]) or etherification (with compounds containing epoxides and halides, such as EPI [59,60,97,142,143,153], BDE [145], glycidyl methacrylate [83], 2,3-epoxypropyltrimethylammonium chloride [154,155,156] and 3-chloro-2-hydroxypropyltrimethylammonium chloride [97]). Surprisingly, with current advances and knowledge in organic chemistry, there is enormous versatility in obtaining (multi)functional starch-based materials. For example, Hu et al. [69] esterified starch with 4-azidobenzoic acid (Figure 2), and then promoted an azide-alkyne click reaction to foster crosslinking with the copolymer of *N*,*N*-dimethylacrylamide and acrylamide modified with alkyne groups, which were also introduced through a click reaction between the amino group of propargylamine and the anhydride moieties of the copolymer. Starch acetylation [157], oxidation to dialdehyde starch [54,55,56,57] or carboxymethylation to CMS [50,51,52,53] are also chemical modifications promoted to improve undesirable properties of native starches. Consequently, the formyl groups of dialdehyde starch or the carboxyl groups of CMS can also react with amines to form Schiff bases [54,57] and amides [53], respectively—Figure 2.

#### 3.1.9. Hydrothermal Carbonization and Carbonization/Pyrolysis

Hydrothermal carbonization (pre-carbonization) and carbonization/pyrolysis consists of relatively cost-effective processes, in which a solid organic compound is subjected to thermal decomposition at high temperatures in a poor oxygen atmosphere, in order to release volatile substances and form a carbon-rich solid residue (char), typically more advantageous in terms of strength-to-weight ratio and with an improved interface, presenting greater affinity and/or selectivity for the sorption of analytes [158,159].

Hydrothermal carbonization is a thermochemical transformation carried out at (180–250) °C in an aqueous medium. Polysaccharides, such as starch, are converted into glycosidic units and dehydration derivatives (organic acids and furfurals) [158]. Carbonization/pyrolysis involves heating at *T* ≥ 350 °C, without any solvent [92,159,160].

For instance, Deng et al. [92] carbonized gelatinized starch at 400 °C, and reacted the starch-based char with peat extract and (2-(dodecyldimethylammonio)acetate (BS) or SDS surfactant to produce efficient multi-modified/carbonized/gelatinized starch adsorbents for Cd(II) and hymexazol fungicide removal. Liang et al. [161] prepared a starch-based carbon by a two-stage hydrothermal reaction at 210 °C, and then promoted carbonization at 800 °C to obtain porous S-PC with high content of oxygen and efficiency for Cr(VI) removal. Moreira et al. [138] first carbonized a material obtained by crosslinking tannin and starch with citric acid (using glycerol as plasticizer) at 550 °C, functionalized it with FeNO_3_ and then promoted a new thermal treatment at 550 °C to obtain Fe@CF for herbicide removal. Zhang et al. [162] prepared magnetic functionalized carbon microspheres (MF-CMS) for the effective removal of tetracyclines by hydrothermal carbonizing starch-rice waste at 230 °C, followed by hydrochar activation with ZnCl_2_ and FeCl_3_, and pyrolysis at 800 °C. Wang et al. [89] also performed a first carbonization of a starch-based polyurethane mixed with K_2_CO_3_ for chemical activation at 500 °C, and then heated it at 800 °C to produce starch polyurethane-activated carbon, STPU-AC, for bisphenol-A adsorption. Furthermore, 3D porous *N*-doped starch-based carbon obtained after carbonization at 550 °C and then 900 °C [163] and Fe-Mn bimetallic doped starch-based porous carbons obtained after heating at 700 °C [101] proved effectiveness for Hg(II) adsorption; while *N*-doped hierarchical porous carbon produced using starch/ammonia as carbon/nitrogen source and carbonized at 800 °C under argon protection was efficiently applied for U(VI) electrosorption [86].

### 3.2. Properties

The properties of pre- and post-adsorption starch-based materials can be studied using various techniques, including Fourier transform infrared spectroscopy (FTIR), Raman spectroscopy, absorption and emission spectroscopy, thermogravimetric analysis (TGA), dynamic light scattering (DLS) and ζ-potential, N_2_ adsorption-desorption isotherms (to determine the Brunaeur-Emmett-Teller surface area (*S*_BET_) and the porosity of the materials), scanning electron microscopy (SEM), energy dispersive X-ray (EDX) mapping, elemental analysis, transmission electron microscopy (TEM), atomic force microscopy (AFM), X-ray diffraction (XRD), X-ray photoelectron spectroscopy (XPS), and vibrating sample magnetometry (VSM). The assessment of the adsorbent before and after adsorption is a key point to understand the processes and mechanisms of the adsorption of pollutants from water using starch-containing bio-based adsorbents with improved performance, together with the effect of experimental parameters (pH, contact time, concentration of pollutants, temperature, interferents, and material regeneration) on adsorption, the analysis of the isotherm, kinetic and thermodynamic of adsorption, and the results of computational simulations, as will be described in Section 4.

Starch-based materials can have different properties depending on the type of modification and the strategy used to achieve it. In general, the preparation of various starch-based formulations aims to overcome the unsatisfactory properties of native starch for environmental remediation application, such as low surface area, molecular weight, thermal stability and variety of reactive functional groups, as well as rapid degradability in water, which greatly limit its adsorption potential [45,164]. Table 1, Table 2 and Table 3 (Section 4) show a comparative analysis of various starch-based materials, regarding its synthesis method and physicochemical characteristics, such as surface area (*S*_BET_), pore size ($d_{P}$), pore volume ($V_{P}$), and pH of zero charge point (pH_PZC_), which are essential to understand the performance of the materials and their interaction with pollutants.

The relationship between structure, properties, and application is very important in the search for high performance materials for adsorption, including those based on starch. Analysing the best performing materials described in Table 1, Table 2 and Table 3, they have thermal stability up to 150 °C [54,92,138] or more [51,52,56,71,81,82,84,88,93,114,138,142,151] (results not reported in Table 1, Table 2 and Table 3), distinct surface area values, ranging from 0.42 to 3262 m^2^ g^−1^, and are generally mesoporous ($d_{P}$ = 2–50 nm), with some exceptions, such as the microporous (3-chloro-2-hydroxypropyl) tri-methyl ammonium chloride-quaternized starch ($d_{P}$ = 1.8 nm) [154], and the following macroporous materials: starch-impregnated MgAl layered double hydroxide [165]; EPI-crosslinked starch and formaldehyde-crosslinked gelatin [144]; ZnO/[starch-poly(vinyl alcohol)] hydrogel with an interconnected porous structure [67]; and chitosan/sodium octenylsuccinate starch aerogel synthesized using high internal phase pickering emulsions as removable templates [70].

Functional optimization of materials also helps to improve their properties and performance. This can be achieved combining starch/modified starch with other materials, such as graphene oxide [53,63,166], metallic particles [25,51,55,64,67,77,85,97,100,101,120,135,138,140,141,153,160,162,165,167,168,169,170], poly(vinyl alcohol) [63,64,67,74,166], polyacrylamide [65], polyethyleneimine [153], poly(acrylic acid) [51,54,65,115], TiO_2_ [103,171], clays [76,114,171], cellulose/modified cellulose [66,85,151], poly(α-L-lysine) [69], pumice [74], tobermorite [84], gelatine [144], alginate [71], chitosan/chitosan derivatives [56,70,150], zeolites [120], and MOFs [88]. Typically, materials with high surface areas present better adsorption capacities, as they have a larger interface and contact area with greater availability of active sites for interaction with analytes. The specific pore volume and the suitability of the pores for the penetration of molecules are also important for ensuring a high adsorption performance. The pH_PZC_ of an adsorbent is another relevant parameter, providing information on the effect of pH on the surface charge of the material, which, depending on the nature and speciation of the adsorbate molecules, can influence the establishment of electrostatic interactions [172].

The assessment of material biodegradability and the analysis of the cost-benefit ratio are also two important characteristics to consider when developing starch-based materials. Some authors have carried out case studies and found that the degradation rates of biopolymers vary according to the modification strategy used to synthesize the starch-based materials. In fact, Ahmad et al. [23] found slower degradation rates for the hydrogel modified with gum tragacanth (28 days) compared to the unmodified hydrogel (21 days), revealing that the gum influenced the degradation behavior. Jumnong et al. [67] synthesized starch-based hydrogels with 50% (*w*/*w*) of biodegradable polymers and found that after 70 days in soil there was good degradation of the biopolymers by microbial action, enzyme-catalyzed hydrogenation, and hydrolysis. Srikaew et al. [111] also found high biodegradation rates for UV-photopolymerized starch-based hydrogels after 60 days in soil. In addition to the high content of biodegradable polymers, the high specific surface area of the materials also favors increased biodegradation rates, due to the action of a greater number of bacteria.

The cost-benefit assessment of the adsorbent includes raw materials, chemicals, and energy costs. For large-scale production, simple, and economical methods and low-cost materials may be employed. The use of starch-based materials can minimize process costs by utilizing a low-cost and widely available biopolymer. Despite this, the synthesis and modification strategies adopted to increase the adsorption capacity of a given adsorbate can increase costs. In this sense, the percentage of modification and its relationship in terms of benefit to the final performance of the material must be studied rigorously to avoid making the material too expensive. This was pointed out by Lencina et al. [71] in the preparation of starch/alginate gels by the external gelation method, which mentioned that the increase of the starch/alginate ratio (by decreasing the alginate content) resulted in a cheaper adsorbent. Moreira et al. [138] stated that replacing the usual crosslinkers by glycerol and citric acid can help to reduce the cost of the process and the level of toxicity of the material. Findik et al. [120] also described coprecipitation as an economically viable alternative for the preparation of magnetic nanoparticles compared to their commercial acquisition.

To get further insights about economic viability, some authors have estimated the cost of material production. For instance, Zhang et al. [162] estimated a cost of 5.91 USD/kg to produce magnetic functionalized carbon microsphere (MF-CMS) and that, per kg of adsorbent, 8 kg of waste rice are consumed and 0.55 tons of tetracyclines (TCs) can be treated in wastewaters. Additionally, desorption of TCs was achieved using 0.1 M NaOH and MF-CMS was regenerated using methanol and reused over 5 cycles, showing a removal rate decrease of only 20% and excellent stability and environmental safety. So, the cost of MF-CMS was comparable to commercial materials (such as activated carbon), however, the traditional adsorbents, besides the high cost, are difficult to regenerate and show a significant reduction in adsorptive capabilities after continuous regeneration cycles [162,173]. Furthermore, it should be noted that most studies are limited to laboratorial experiments, and little research has been carried out using real wastewater samples and scale-up.

## 4. Application in Contaminant Removal

According to the United Nations, worldwide, the use of water resources has grown twice as fast as population growth [174]. Due to water supply systems surrounding agricultural, domestic, or industrial environments, rapid urbanization in recent years has increased the contamination of natural ecosystems with potentially toxic pollutants (e.g., heavy metals and emerging pollutants [175]). Thus, several removal technologies can be employed to mitigate these pollutants, highlighting bioremediation, ion exchange, adsorption, coagulation, flotation, membrane filtration, and chemical precipitation [176]. Among them, promising attention has been given to adsorption strategies using different materials based on biopolymers (including starch), activated carbon, carbon nanotubes, biochar, clays, among others, due to their high efficiency, simplicity, flexibility of operation, and pollutant removal at low concentrations [177,178]. Although starch can be used efficiently in various applications (including food packaging and biodegradable films, drug and fertilizer delivery, emulsion, catalysis [179,180,181,182,183]), nowadays there is a huge opportunity to obtain promising starch-based structures for adsorption in order to mitigate environmental and water pollution, as will be described in the following subsections.

### 4.1. Adsorption of Heavy Metals

Heavy metals are pollutants that have concerned the technical-scientific community, especially Cd(II), Pb(II), Cr(III) and Cr(VI), Cu(II), Hg(II), Co(II), Zn(II), and Ni(II), due to their presence in water in low concentrations and their highly harmful effects on health [184]. These metals are common contaminants in industrial waste from mining, electroplating, alloys and metallurgy, manufacturing of electronic components, and recycling of electronic waste, being capable of causing serious toxicological effects [176,185]. Considering the high industrial demand for metals, their removal/recovery from secondary sources is of environmental and economic interest from the circular economy perspective, through the reuse of resources already extracted in a new production cycle [186,187]. Thus, due to the accumulation of metals in aquatic biota and their market demand, the development of efficient adsorbent matrices to remove and recover these elements from secondary sources has become increasingly important [188]. In this sense, a great diversity of starch-based adsorbents has been prepared through different synthesis methods for heavy metal removal. Table 1 summarizes information on the various starch adsorbents developed since 2023 for metal remediation. Due to the number of promising materials, the works of Wang et al. [160] and Tan et al. [153] were considered as examples.

**Table 1 polymers-17-00015-t001:** Starch-based adsorbents with the best adsorption capacities for each heavy metal removal reported since 2023: synthesis, properties & adsorption parameters, isotherm, kinetic, and thermodynamic.

Adsorbate	Best Starch-Based Adsorbents	Synthesis Method	*S*_BET/_ (m^2^ g^−1^)	$d_{P}$/nm	$V_{P}$/(cm^3^ g^−1^)	pH_PZC_	*R*_S-L_/ (g L^−1^)	pH	*T*/K	Isotherm		Kinetic		Thermodynamic		
										Model	$q_{max}/$(mg g^−1^)	Model	$t_{eq}/$min	${∆G}^{0}$/(kJ mol^−1^)	${∆H}^{0}$/(kJ mol^−1^)	${∆S}^{0}$/(J K^−1^ mol^−1^)
Ni(II)	Blend hydrogels of cellulose nanocrystals and corn starch [151]	Physical crosslinking & free radical polymerization	−	−	−	−	0.01	6.0	300	L	276	PSO	120	−	−	−
	Magnetic *N*-vinylformamide grafted starch [104]	Free radical graft copolymerization, coprecipitation & sonication	22	7.8	0.038–0.049	−	2.00	6.0	298	F	9	−	−	−	−	−
Cu(II) [50,51,54,71,77,78,79,82,83,85,99,104,132,134,137,143,189,190]											275					
	Acylhydrazone-functionalized dialdehyde starch/poly(AA-co-methylmethacrylate) based adsorbent [54]	Free radical polymerization, modification & Schiff base crosslinking	36	26.7	−	4.87	1.00	5.0	298	L	411	PSO	50	−34 to −30	32	206
	Corn starch/AA/itaconic acid ion exchange hydrogel [132]	Free radical graft copolymerization & crosslinking	−	−	−	−	0.40	4.5	298	L	699	PSO	40	−11 to −7	50	190
Pb(II) [50,51,52,57,58,77,102,132,142,160,189,191]							0.30				870			−14 to −9	68	258
	Poly(AA)/carboxymethyl starch/Fe_3_S_4_ magnetic hydrogel [51]	Free radical graft self-polymerization	−	−	−	2.32	3.00	6.0	303	L–F	234	PSO	200	−10 to −2	111	387
	Starch-stabilized and Fe/Ni bimetal modified biochar composite [160]	Pyrolysis, coprecipitation & pyrolysis	184	5.7	0.047	4.25	0.50	5.0	298	L	156	PSO	240	−21 to −15	−30	−0.08
	Starch-coated CuFe_2_O_4_-modified magnetic biochar composites [77]	Pyrolysis, oxidation, coating modification & amino functionalization	221	4.9	0.102	4.31	0.50	5.0	298	L	201	PSO	360	−41 to −40	6	61
Cd(II) [25,75,77,84,92,139,189,191]											191			−41 to −37	20	104
	Drought-resistant and water-retaining tobermorite/starch composite hydrogel [84]	Free radical polymerization & crosslinking	31	11.5	0.180	−	1.00	5.2	298	L	591	PSO	480	−	−	−
	Zn/Fe-bimetal-loaded and starch-coated corn cobs biochar [25]	Pyrolysis, coprecipitation & coating modification	197	-	0.095	4.41	0.50	5.0	298	L	184	PSO	240	−49 to −43	29	139
U(VI)	Esterified sago hampas [90]	Hydrothermal alkalization, acidification, & esterification	−	−	−	−	1.00	4.0	298	S	16	PSO	150	−9 to −3	−99	−0.3
	Nitrogen doped hierarchical porous carbon (NHPC) [86]	In-situ growth & pyrolysis	2000	3.2	1.610	−	−	6.0	298	L	194	−	−	−	−	−
Hg(II)	Three-dimensional porous *N*-doped starch-based carbon [163]	Ni(II) & Mn(II)-catalyzed co-pyrolysis	20	−	0.059	1.60	0.20	5.0	293	F	403	PSO	90	−11 to −7	22	87
	Mn bimetallic doped starch-based porous carbons adsorbents (Fe-Mn@SCAs) [101]	Hydrothermal synthesis & carbonization	26	2.3	−	2.30	0.20	6.0	303	L	325	PSO	90	−4 to −0.1	25	88
	Self-assembled starch-based organometallic polymers rich in oxygen vacancy/sulfhydryl groups [81]	High-pressure solvothermal synthesis	−	−	−	3.40	0.10	6.0	298	L	2306	PSO	30	−	−	−
Cr(III)	Modified carboxymethyl corn starch/graphene oxide composite aerogel [53]	Hydrothermal crosslinking	−	−	−	−	2.00	5.0	298	F	235	PSO	180	−	−	−
	Starch-graft-itaconic acid hydrogels [133]	Crosslinking, free radical polymerization & graft	−	−	−	4.00	2.00	8.0	298	−	14	−	−	−	−	−
Cr(VI) [80,104,133,137,153,161]	Starch-based polyporous carbon [161]	Two step hydrothermal carbonization & pyrolysis	1430	2.6	0.920	5.10	0.40	2.0	298	F	884	PSO	480	−12 to −11	−57	244
	Magnetic starch integrating Fe_3_O_4_, modified with EPI and polyethyleneimine [153]	Etherification & crosslinking	3.6	<10.0	0.001	>11.00	−	7.0	298	L	165	PSO	60	−46 to −40	0.03	144
	Magnetic *N*-vinylformamide grafted starch [104]	Free radical graft copolymerization, coprecipitation & sonication	22	7.8	0.038–0.049	−	2.00	6.0	298	F	37	−	−	−	−	−
Co(II)	Starch-AA-nanohalloysite composite [114]	γ-rays induced synthesis	−	−	−	2.10	2.00	5.0	298	L	105	PSO	150	−	−	−
	Starch-grafted citric acid-acrylamide/magnesia hydrogel [140]	Free radical polymerization	−	−	−	−	−	7.0	298	L	113	PSO	40	−	−	−
Cs(I)											100					
Ag(I)	Poly(AA) grafted carboxymethyl chitosan/dialdehyde starch [56]	Schiff base modification, free radical polymerization & crosslinking	14	2–8	−	−	12.00	-	313	L	503	PSO	30	−	−	−
Sr(II)	Bismuth doping starch-poly[2-(dimethylamino) ethyl methacrylate] hydrogel [91]	γ-rays induced synthesis	−	−	−	−	−	6.0	303	L	59	PSO	180	−1.9 to −1.8	>0	>0
Zn(II)	Crosslinked carboxymethyl starch-g-methacrylic acid (CCMS-g-MAA) [52]	Free radical graft polymerization & crosslinking	−	−	−	<3.00	0.70	5.0	293	F	51	PSO	70	−	−	−
Fe(II)	Biobased biodegradable hydrogel containing modified cellulosic nanofiber-ZnO nanohybrid [85]	Free radical graft polymerization & crosslinking	1.0	3.8	0.002	−	0.10	7.0	303	F	70	PSO	30	−	−	−
	Starch and chitosan biopolymer blend [150]	Physical crosslinking	4.0	−	−	6.50	3.33	5.6	298	L	115	PSO	960	−	−	−

*S*_BET_: Brunaeur-Emmett-Teller surface area; $d_{P}$: pore size; $V_{P}:$ pore volume; pH_PZC_: pH of zero charge point; *R*_S-L_: solid-liquid ratio; *T*: temperature of the adsorption assays; $q_{max}$: maximum adsorption capacity; $t_{eq}$: equilibrium time; ${∆G}^{0}$: Gibbs free energy of adsorption; ${∆H}^{0}$: enthalpy of adsorption; ${∆S}^{0}$: entropy of adsorption; L: Langmuir; F: Freundlich; PSO: pseudo-second order.

Wang et al. [160] developed starch-stabilized Fe-Ni/biochar composites (StFeNi-BC and StFeNi-BC-350) to improve Pb(II) removal from wastewater and reduce Pb(II) stress in soil-wheat pot systems. The incorporation of Fe/Ni bimetal and starch into biochar (BC) resulted in composites with higher specific surface area (184 m^2^ g^−1^ for StFeNi-BC-350, 157 m^2^ g^−1^ for StFeNi-BC, 87 m^2^ g^−1^ for FeNi-BC, and 5 m^2^ g^−1^ for BC), larger average crystallite sizes (4.7, 16.0 and 21.4 nm for FeNi-BC, StFeNi-BC and StFeNi-BC-350, respectively), and smaller pore sizes and volumes (5.7 nm and 0.047 cm^3^ g^−1^ for StFeNi-BC-350, 7.5 nm and 0.062 cm^3^ g^−1^ for StFeNi-BC, 10.5 nm and 0.121 cm^3^ g^−1^ for FeNi-BC, and 13.5 nm and 0.124 cm^3^ g^−1^ for BC, respectively). The calcination of StFeNi-BC to StFeNi-BC-350 at 350 °C for 3 h resulted in a better Pb(II) adsorption performance. The pH of the medium was shown to be important in the adsorption capacity by affecting the metal speciation, surface charge distribution of the adsorbents, and the dissociation of functional groups. Therefore, the adsorption capacity of Pb(II) gradually increased with the increase of pH and tended to decrease at pH = 5–6, so the best performance was obtained at pH = 5 (Figure 3A). This trend occurs since the positive surface charge of the materials (due to the dissociation of the hydroxyl/carboxyl groups on the adsorbent surface) decreases until become negative at pH > pH_PZC_ (pH_PZC_ = 6.9 (BC), 4.7 (FeNi-BC), 4.4 (StFeNi-BC), and 4.3 (StFeNi-BC-350)), thus increasing the contribution of the attractive electrostatic effects. The SEM-EDX spectrum of StFeNi-BC-350 surface after Pb(II) adsorption also proved the success of Pb(II) adsorption (Figure 3B).

The best description of the kinetic data (Figure 3C) by the pseudo-second order (PSO) model confirmed the occurrence of chemisorption by ionic bonds. Additionally, the fitting of the intraparticle diffusion model identified three steps in Pb(II) adsorption: (i) the adsorbate diffusion to the external surface of the adsorbent; (ii) the adsorbate diffusion through the composite pores; and (iii) the equilibrium adsorption (Figure 3D). The best fit of the equilibrium data to the Langmuir (L) model indicated a homogeneous monolayer adsorption system, with maximum adsorption capacities ($q_{max}$) varying between 78 (BC) to 156 mg g^−1^ (StFeNi-BC-350), being StFeNi-BC-350 a good adsorbent for Pb(II) removal (Table 1). The capture of Pb(II) by the composites proved to be a spontaneous (${∆G}^{0}$ < 0, between −21 and −15 kJ mol^−1^), exothermic (${∆H}^{0}$ < 0, equal to −30 kJ mol^−1^) and entropy-decreasing (${∆S}^{0}$ < 0, equal to −0.08 J K^−1^ mol^−1^) process and to have good reusability, since starch-based composites maintained more than 80% of the initial adsorption capacity after 5 cycles (Figure 3E). As the reuse cycles increased, the adsorption performance gradually decreased. This might be due to the loss of adsorption sites of the adsorbent during regeneration. Furthermore, the authors highlighted the relative contribution of different mechanisms to Pb(II) removal: the complexation with COOH and OH functional groups was predominant (45%), followed by electrostatic attraction (26%), ion exchange with Ca(II) and K(I) (18%) and physical adsorption (11%). It was also observed that StFeNi-BC-350 presents a high anti-interfering ability in the presence of co-existing metal ions (Na(I), K(I), Cr(VI), Mg(II), Ca(II), Co(II), Cd(II)) (Figure 3F), maintains its performance in tap and river water, but undergoes a performance decrease in industrial wastewater due to competitive binding. Additionally, in soil-wheat pot systems, StFeNi-BC-350 increased the biomass and total chlorophyll content in leaves, and decreased the activity of antioxidant enzymes together with fluorescence intensity of wheat, meaning that it can alleviate Pb(II) oxidative stress of wheat [160].

Regarding Cr(VI) oxyanions adsorption, the magnetic starch composite (MAST) obtained by Tan et al. [153] can be highlighted. For its synthesis, the hydroxyl groups in the amylopectin chain underwent an etherification reaction with EPI to introduce epoxy groups. Subsequently, polyethyleneimine was introduced as catalyst for the crosslinking reaction, leading to the incorporation of amino groups in the starch structure, which reacted with epoxy groups in other sites of the macromolecule. Magnetic Fe_3_O_4_ nanoparticles were combined with starch before the previously described reactions. The mesoporous MAST, presenting a surface area of 3.6 m^2^ g^−1^, $d_{P}$ < 10 nm, and $V_{P}$ = 0.001 cm^3^ g^−1^, revealed exceptional maximum adsorption capacities for various hazardous substances, including diclofenac (621 mg g^−1^), methyl orange (471 mg g^−1^), amaranth (194 mg g^−1^), and Cr(VI) (165 mg g^−1^). The adsorption data, described by the Langmuir model, suggested that all adsorbates were captured at specific sites on the surface of MAST, resulting in monolayer adsorption. Thermodynamic parameters showed that all adsorption processes were spontaneous (${∆G}^{0}$ = −46 to −40 kJ mol^−1^), endothermic (${∆H}^{0}$ = 0.03 kJ mol^−1^, with increased efficiency at higher temperatures), and resulting in a random entropy increase (${∆S}^{0}$ = 144 J K^−1^ mol^−1^). MAST achieved rapid equilibrium within 60 min and the best fit of the PSO model suggested that adsorption was governed by surface reaction limitations. Notably, the composite showed stable adsorption capacity in the presence of various organic mixtures (acetic acid, ethanol, sucrose), pH values, sodium salts, and interfering cations, also exhibiting selectivity for anionic species (e.g., Cr(VI) oxyanions), since it has a positive surface charge, even at pH = 11. Furthermore, MAST was easily separated from water using a magnet, and retained a Cr(VI) removal efficiency above 60% even after five reuse cycles, endorsing its feasibility for reuse. However, an increase in the number of adsorption cycles can lead to a decline in adsorption capacity, which could be attributed either to the saturation of active sites on the adsorbent’s surface or partial damage of the adsorbent structure, resulting in a gradual reduction in MAST adsorption capacity. Thus, the choice of the conditions to carry out adsorption/desorption cycles must be fine-tuned to minimize structural changes, mass losses, and leaching/inactivation of functional groups, to achieve quantitative desorption efficiencies when adsorption is reversible, and to ensure that the material maintains its performance over many cycles (which will be important for minimizing secondary pollution caused by its disposal). Reaching all these conditions simultaneously is a huge challenge, since the optimization of the operation parameters is performed for each individual system and involves a compromise between the different variables.

The mechanism of adsorption of Cr(VI) on MAST was investigated by SEM-EDX and XPS analyses. The SEM-EDX mapping (Figure 4A–C) showed the presence of C, O, N, Cl (from EPI), K (from potassium dichromate), Fe (from Fe_3_O_4_), and Cr, confirming the effectiveness of Cr(VI) adsorption, which was further evidenced by the point scan analysis of the elemental distribution using the EDX spectrum (Figure 4D). In XPS analysis, both the XPS survey (Figure 4E) and the Cr 2p (Figure 4F) spectra of MAST after Cr(VI) adsorption allowed distinguishing peaks corresponding to Cr(VI) (Cr 2p_3/2_) and Cr(III) (Cr 2p_1/2_), confirming that a portion of the adsorbed Cr(VI) was reduced to Cr(III). The adsorption/desorption behaviors of MAST at different pH levels indicated that the primary mechanism of Cr(VI) adsorption was via electrostatic interactions. However, in desorption experiments, complete removal of adsorbed Cr(VI) was not achieved, probably due to ion exchange and more rigid interactions, such as hydrogen bonding, and van der Waals forces that also occur during adsorption, after different forms of Cr(VI) oxyanions (Cr_2_O_7_^2−^, CrO_4_^2−^, HCrO_4_^−^) predominantly adsorbed by electrostatic attraction enter the interior of MAST adsorbent.

### 4.2. Adsorption of Emerging Pollutants

Emerging pollutants are contaminants of current concern for which there is no regulation on the maximum acceptable values in water, although they can affect the environment and living beings, even at low concentrations [192,193]. These include endocrine-disrupting chemicals, pesticides, pharmaceuticals, hormones, toxins, micro/nanoplastics, synthetic dyes from industrial sources and dye-containing dangerous contaminants [194,195,196]. Since 2023, many starch-based materials have been synthesized and applied to remove compounds from this class of pollutants—Table 2 and Table 3, for dyes and other emerging pollutants, respectively. Particularly for dye adsorption, it is worth noting that a wide variety of materials have proved to be efficient for the remediation of methylene blue, crystal violet, malachite green and methyl violet cationic dyes, and methyl orange and congo red anionic dyes. For these cases, Table 2 lists the three materials that presented the highest maximum adsorption capacity ($q_{max}$) for each dye.

Acylhydrazone-functionalized dialdehyde starch/poly(AA-co-methylmethacrylate) based biosorbent (Figure 5A) with an abundant 3D pore-like structure (abbreviated as GSL) was prepared by a Schiff base crosslinking reaction between the formyl groups of dialdehyde starch and the NH_2_ groups of poly(AA-co-methylmethacrylate) modified with hydrazine. For this synthesis, poly(AA-co-methylmethacrylate) was previously obtained through a hydrothermal free radical polymerization involving methyl methacrylate, AA, MBA, and KPS as initiator. The mesoporous GSL, presenting a surface area of 36 m^2^ g^−1^, exhibited remarkable $q_{max}$ of 2102 and 2237 mg g^−1^ for malachite green (MG) and safranin O (SO) cationic dyes at pH = 6.0 and 25 °C. As the pH increased to 5–6, the adsorption performances increased and then remained constant, possibly due to the occurrence of electrostatic interactions between the cationic dyes and the negatively charged surface of GSL (pH_PZC_ = 4.9). For both adsorbates, the adsorption equilibrium and kinetics were described by Langmuir and PSO models, respectively, suggesting monolayer chemisorption mechanisms. The fits to the intraparticle diffusion and film diffusion kinetic models did not pass through the (0,0) point, showing the importance of both intraparticle and liquid film diffusion. Thermodynamically, adsorption processes are spontaneous (${∆G}^{0}$ = −37 to −32 kJ mol^−1^ (SO) and −35 to −30 kJ mol^−1^ (MG)), endothermic (${∆H}^{0}$ = 45 kJ mol^−1^ (SO) and 20 kJ mol^−1^ (MG)), and lead to an increase in entropy (${∆S}^{0}$ = 257 J K^−1^ mol^−1^ (SO) and 172 J K^−1^ mol^−1^ (MG)), suggesting a growth of the disorder at the solid/liquid interface. In this work, continuous-flow adsorption of SO was also performed, and it was observed that the adsorption capacity increased with a lower flow rate, and higher adsorbent mass and initial pollutant concentration (*C*_0_) (Figure 5B, Figure 5C and Figure 5D, respectively). An increase in the concentration or adsorbent mass and a decrease in the flow rate have also led to a decrease in the kinetic constants. Under the best conditions (0.7 g of GSL, 0.48 mL min^−1^ flow rate, *C*_0_ = 1400 mg L^−1^), a column saturation capacity of 1101 mg g^−1^ for SO was obtained [54]. Continuous adsorption studies were also carried out as a scale-up strategy using other starch-based materials [169,170,197,198].

To better understand the mechanism of adsorption of MG and SO on GSL, FTIR, XPS and SEM analyses were carried out. According to the first technique, shifts and an intensity decrease in O–H elongation bands on GSL after adsorption suggest the involvement of OH groups in hydrogen bonds with the adsorbates, while shifts in C=O peak can be related to the occurrence of electrostatic attraction and n–π interactions. In XPS, changes were observed in C 1s and O 1s spectra for GSL after dye adsorption, namely deviations in the binding energies of C–O, C–OH and C=O, O–C=O bonds, due to their relevance in electrostatic, hydrogen and n–π interactions with both dyes (Figure 5E–H for SO adsorption) [54]. In SEM analysis after adsorption, less porous, and denser or rougher surfaces were observed, proving adsorption, as pollutant species can cover the surface and fill the pores. Figure 5I also shows that GSL can be applied efficiently for Cu(II) and sulfide removal, and maintains its high sorption ability over 6 reuse cycles, and then loses some of the removal capacity for all pollutants until the 15th cycle. Additionally, the performance of GSL remained practically unchanged in well water, lake water, river water and tap water; however, in real industrial wastewater, its adsorption efficiency for MG, SO, Cu(II) and sulfides decreased due to interference from other organic and inorganic compounds [54].

**Table 2 polymers-17-00015-t002:** Starch-based adsorbents with the best adsorption capacities for each dye removal reported since 2023: synthesis, properties & adsorption parameters, isotherm, kinetic, and thermodynamic.

Adsorbate	Best Starch-Based Adsorbent	Synthesis Method	*S*_BET/_ (m^2^ g^−1^)	$d_{P}$/nm	$V_{P}$/(cm^3^ g^−1^)	pH_PZC_	*R*_S-L_/(g L^−1^)	pH	*T*/K	Isotherm		Kinetic		Thermodynamic		
										Model	$q_{max}$/(mg g^−1^)	Model	$t_{eq}$/min	${∆G}^{0}$/(kJ mol^−1^)	${∆H}^{0}$/(kJ mol^−1^)	${∆S}^{0}$/(J K^−1^ mol^−1^)
Acid red-97	Graphene oxide/starch–PVA hydrogel [63]	Hydrothermal crosslinking	–	–	–	5.00	1.00	2.0	303	F	8	PSO	90	−5 to −2	−57	172
Golden yellow-160	Zn-NiFe_2_O_4_/starch composite [169]	Hydrothermal crosslinking	–	–	–	6.00	0.60	3.0	308	L	47	PSO	90	−1 to 1	16	52
Reactive red	Hydrogel of PVA & starch crosslinked with boric acid and modified with copper-doped zinc oxide [64]	Hydrothermal crosslinking & coprecipitation	89	~4.0	~3.800	6.70	1.00	3.0	303	L	79	PFO	60	−2 to 2	−37	−116
Methylene blue [23,49,61,62,65,66,67,68,69,70,79,81,93,96,106,110,111,129,130,141,152,165,166,171,197,199,200,201,202,203,204,205,206,207,208,209,210]	CMS-co[polyacrylamide/poly(AA)] hydrogel [65]	Free radical graft polymerization & crosslinking	–	–	–	<3.00	0.89	7.0	298	L	1700	PSO	120	–	–	–
	Starch/carboxymethylcellulose hydrogel grafted with MBA, AA, 2-acrylamido-2-methylpropane sulfonic acid [66]	Free radical graft polymerization & crosslinking	–	–	–	–	0.10	5.1	298	L	1625	PSO	60	−10 to −8	−43	−110
	ZnO/[starch-PVA] hydrogel [67]	Free radical graft polymerization & precipitation	–	15,000–40,000	–	6.50	5.00	–	r.t.	L	1170	PSO	20	−6 to −1	41	148
Amaranth	Magnetic starch integrating Fe_3_O_4_, modified with EPI and polyethyleneimine [153]	Etherification & crosslinking	3.6	<10	0.001	>11.00	–	7.0	298	L	195	PSO	60	−50 to −43	0.03	152
Methyl orange [103,153,156,166,211,212]											474			−55 to −46	0.04	165
	Starch/PVA/graphene oxide nanocomposite [166]	Hydrothermal, sonication & casting methods	97	3.5	–	5.80	0.40	5.0	298	L	293	PSO	150	−4	−14	−32
	TiO_2_/starch-based microparticles [103]	Microwave, modification & precipitation	–	–	–	–	–	–	298	L	547	PSO/ PFO	40	–	–	–
Crystal violet [57,69,73,103,141,171,208,213,214,215,216]											856					
	Oxidized starch modified with histidine [57]	Oxidation & Schiff base modifications	–	–	–	–	2.00	5–6	r.t.	F	248	PSO	30	−50 to −49	−26	−77
	Starch modified red clay [171]	Hydrothermal modification & calcination	–	–	–	–	1.00	–	r.t.	L	161	PSO	60	–	–	–
Acid blue 41	Zn–Al layered double hydroxide-manganese ferrite–eggshell biochar–starch [170]	Carbonization & coprecipitation	–	–	–	4.00	1.00	2.0	303	F	40	PSO	60	−4 to 0	−40	−121
Brilliant green	ZnFe_2_O_4_-starch glutaraldehyde-crosslinked composite [141]	Crosslinking	3.0	45.4	3.679	7.40	2.00	7.0	298	F	96	PSO	90	−5 to 0	13 to 92	0.061 to 0.312
Basic violet 7	Polyvinyl alcohol/starch/pumice composite films [74]	Hydrothermal crosslinking, casting & irradiation	–	–	–	–	4.00	11.0	298	F	65	PSO	420	−17 to −16	9	54
Acid black	*N*-2,3-epoxypropyl trimethyl ammonium chloride-based cationic starch [156]	Hydrothermal acid hydrolysis & solvothermal etherification	–	–	–	–	2.00	–	–	–	237	PSO	160	–	–	–
Safranin O	Acylhydrazone-functionalized dialdehyde starch/poly(AA-co-methylmethacrylate) based adsorbent [54]	Free radical polymerization, hydrazide modification & acylhydrazone-forming crosslinking	36	26.7	–	4.87	1.00	6.0	298	L	2237	PSO	160	−37 to −32	45	257
Malachite green [54,68,70,115,131,208]											2102			−35 to −30	20	172
	Starch-grafted poly(AA) copolymer with acrylamide [115]	Free radical graft copolymerization	–	–	–	5.20	0.67	8.0	298	L	160	PSO	30	−17 to −10	52	208
	Starch, itaconic acid, AA-based hydrogel containing modified cellulose nanofiber [68]	Free radical graft copolymerization & crosslinking	–	–	–	–	0.20	–	r.t.	F	405	PSO	40	–	–	–
Cresol red											323		80			
Eosin Y	Hydrogel of azide-modified starch/poly(α-L-lysine)/alkyne-modified copolymer of *N*,*N*-dimethylacrylamide and acrylamide [69]	Esterification & amino-anhydride, azide-alkyne crosslinking reactions	0.73	6.4	0.002	4.20	0.40	4.0	298	L	106	PSO	120	−23 to −20	53	246
Congo red [23,55,69,210]							–	–		–	100	–	120	–	–	–
	Biochar of enzyme-undigestible residues from starch-containing rice straws [210]	Carbonization	3262	~2.5	1.720	–	1.00	–	298	L	2714	–	360	–	–	–
	MXene immobilized with dialdehyde starch nanoparticles [55]	Hydrothermal crosslinking	7.9	9.2	0.029	<2.00	0.10	7.0	303	L	754	PSO	700	<0	<0	<0
Rhodamine B											678					
Acridine orange	Chitosan/sodium octenylsuccinate starch aerogel [70]	Emulsification, UV polymerization & freeze-drying	–	43,540	–	–	0.67	–	–	F	79	PSO	375	–	–	–
Methyl violet [68,70,120,217]											52					
	Starch, itaconic acid, AA-based hydrogel containing modified cellulose nanofiber [68]	Free radical graft copolymerization & crosslinking	–	–	–	–	0.20	–	r.t.	F	377	PSO	40	–	–	–
	Zeolite/MgO/Starch/Fe_3_O_4_ magnetic nanocomposite [120]	Coprecipitation	34	–	–	3.50	4.00	5.8	295	F	7	PSO	45	–	–	–
Direct black 22								7.5		L	4					
Methyl blue	(3-chloro-2-hydroxypropyl) tri-methyl ammonium chloride-quaternized starch [154]	NaOH-catalyzed etherification	41	1.8	0.036	6.40	1.00	2.0	303	–	263	PFO	75	–	–	–
Acid violet 19	Low-cost Brazilian corn starch [218]	–	–	–	–	–	–	7.0	–	–	–	PSO	60	–	–	–
Direct dye 4BS	2-methacryloyloxyethyl trimethyl ammonium chloride & dimethyl diallyl ammonium chloride-based cationic starch [136]	Free radical graft copolymerization	15	5–30	–	9.08	2.50	6.0	298	L	212	PSO	30	−18 to −17	−11	92,870
Reactive green KE-4B											111		60	−21 to −19	−29	157,180
	Straw-reinforced grafted starch composite resin through copolymerization of dimethyl diallyl ammonium chloride, 2-methacryloxyethyl trimethylammonium chloride [198]	Graft & crosslinking copolymerization	17	5–15	–	9.21	5.00	3.0	298	–	~16	–	120	–	–	–
Reactive red X-3B											~19					
Direct blue 5B											~18					
Reactive blue 19	Starch modified NiFe layered double hydroxide composites [100]	Coprecipitation	6.7	3.4	0.011	–	0.50	6.3	r.t.	L	149	PSO	180	–	>0	–
Reactive orange 16											182			–		–

PFO: pseudo-first order.

**Table 3 polymers-17-00015-t003:** Starch-based adsorbents for other emerging poluttants (apart from dyes) removal reported since 2023: synthesis, properties & adsorption parameters, isotherm, kinetic, and thermodynamic.

Adsorbent	Synthesis Method	*S*_BET/_ (m^2^ g^−1^)	$d_{P}$/nm	$V_{P}$/(cm^3^ g^−1^)	pH_PZC_	Adsorbate	*R*_S-L_/(g L^−1^)	pH	*T*/K	Isotherm		Kinetic		Thermodynamic		
										Model	$q_{max}$/(mg g^−1^)	Model	$t_{eq}$/min	${∆G}^{0}$/(kJ mol^−1^)	${∆H}^{0}$/(kJ mol^−1^)	${∆S}^{0}$/(J K^−1^ mol^−1^)
Microplastics and nanoplastics																
Ultra-light sponge of EPI-crosslinked starch and formaldehyde-crosslinked gelatin [144]	Hydrothermal crosslinking	–	10^5^	–	–	Microplastics	1.25	7.0	293	–	22	PFO	1320	–	–	–
						Nanoplastics					20					
Pharmaceuticals																
Montmorillonite/MnFe_2_O_4_/Starch/ZIF-67 [76]	Coprecipitation	588	10.9	0.350	8.50	Tetracycline	0.50	6.0	298	L	135	PSO	60	−12 to −9	−48	−122
*N*-Fe_3_O_4_ magnetic carbon microsphere [167]	Hydrothermal & carbonization	383	3.4	0.466	5.80		1.00	5.0	r.t.	L	111	PSO	120	–	–	–
Magnetic functionalized carbon microsphere (MF-CMS) [162]	Hydrothermal carbonization & carbonization	748	3.4	0.405	–		0.50	5.0	298	L	95	PSO	360	–	>0	–
						Doxycycline					68					
						Oxytetracycline					80					
						Chlortetracycline					65					
EPI-crosslinked cationic starch microgranule [155]	Crosslinking & cationization	–	–	–	–	Diclofenac	0.50	6.5	303	L	629	PSO	30	−13 to −12	−17	−16
Magnetic starch integrating Fe_3_O_4_, modified with EPI and polyethyleneimine [153]	Etherification & crosslinking	3.6	<10.0	0.001	>11.00		–	7.0	298	L	621	PSO	60	−57 to −48	0.06	170
EPI-crosslinked starch ester [59,60]	Crosslinking	0.43	–	1.561	~7.00	Ciprofloxacin	2.00	7.0	298	L	244	PFO	180	−4 to −3	−17	−45
Starch-impregnated MgAl layered double hydroxide [165]	Coprecipitation	37	323	0.243	–	Amoxicillin	0.05	7.0	298	L	48	PSO	120	−15 to −14	−41	−85
Molecularly imprinted polymer incorporated with starch capped silver nanoparticles [135]	Solvothermal free radical polymerization	–	–	–	–	Sulfamethoxazole	–	7.0	298	L	31	PSO	40	−11 to −9	8	55
						Nevirapine					37		30	−13 to −10	11	70
						Ibuprofen					30		50	−14 to −10	14	79
EPI-crosslinked cationic potato starch granules containing CaO [97]	Crosslinking, cationization & sonochemistry	–	–	–	–	Ibuprofen	0.50	–	298	L	555	PSO	30	–	–	–
MXene immobilized with dialdehyde starch nanoparticles [55]	Hydrothermal crosslinking	7.9	9.2	0.029	<2.00	Naproxen	0.10	7.0	303	L	167	PSO	240	<0	<0	<0
Starch modified NiFe layered double hydroxide composites [100]	Coprecipitation	6.7	3.4	0.011	–	Piroxicam	0.50	8.3	r.t.	F	2840	PSO	300	–	>0	–
Endocrine-disrupting compound																
K_2_CO_3_ activated carbon derived from starch-based polyurethane [89]	Crosslinking & carbonization	1885	<2.0	0.700	5.30	Bisphenol A	0.50	5–6	298	F	521	PSO	120	−33 to −27	65	310
Pesticides																
Multi-modified with peat extract & BS/carbonized/gelatinized starch [92]	Carbonization & modification	0.42	17.2	0.002	–	Hymexazol	–	5.0	293	F	~14	–	–	<0	<0	>0
Ag@starch-based carbon-dots core-shell nanohybrid material functionalized with 1,2-naphthoquinone-4-sulphonic acid [168]	Hydrothermal carbonization & modification	–	–	–	–	Atrazine	5.00	7.4	r.t.	F	208	–	–	–	–	–
Fe-functionalized rigid carbon foam derived from crosslinked polymer of starch and tannin with glycerol and citric acid [138]	Crosslinking, carbonization & functionalization	318	2.3	0.187	7.35		1.00	7.0	298	L	88	PFO	90	−30 to −27	8	119
						Diuron					36	PSO	180	−29 to −25	39	214
						2,4-D					5	PSO	240	−22 to −21	−20	4
Nanocrystalline Hf(IV)-MOF embedded starch@cotton composite [88]	Hydrothermal synthesis	–	–	–	–	Trifluralin	30.00	–	r.t.	–	179	–	15	–	–	–
						Nanoplastics					20					

Between adsorption cycles, the previous adsorbent (GSL) was chemically regenerated using a 1 mol dm^−3^ HCl solution [54]. In general, chemical desorption using different eluents (acid, bases, chelating agents, salts, organic solvents) is the preferred strategy for the regeneration of starch-based materials and numerous types of novel adsorbents synthesized in recent years [193,219,220]. However, other approaches can also be employed, albeit less frequently, such as thermal regeneration (typically for conventional adsorbents, e.g., activated carbon or zeolites, since starch-containing materials and organic-based adsorbents degrade at aggressive temperature conditions [219,221]) or bioregeneration/biodegradation (i.e., by microorganism activity and enzymatic reactions [222,223]). It is also important to note that economic and environmental issues have to be taken into account when searching for an appropriate desorption method, since regeneration at high temperatures involves high energy costs; while elution can involve unsafe and unsustainable substances, and require additional steps to regenerate (e.g., neutralization) the material before a new adsorption cycle. In this sense, an advantageous approach to explore is the eluent reuse in pollutant desorption and concentration [224].

To fight against other types of emerging pollutants (i.e., pesticides), Moreira et al. [138] proposed a Fe-functionalized rigid carbon foam derived from carbonization of a starch and tannin-based crosslinked polymer, as described in Section 3.1.9. The best foam was obtained without Pluronic F-127 as soft template, using starch/tannin and (starch+tannin)/glycerol ratios of 2.0 and 0.75, respectively, and the lowest iron impregnation content of 8% (*w*/*w*) (Figure 6A). This is in line with the cost-benefit analysis required in the material optimization process (discussed in Section 3.2) and, in this case, the addition of Pluronic F-127 and increased percentages of Fe were not advantageous, making the most efficient material simpler, and more economical and environmentally viable. The resulting Fe@CF presented S_BET_ = 318 m^2^ g^−1^, mesoporous of 2.3 nm and 50% porosity, and showed the following order of affinity for the adsorptive removal of herbicides: atrazine > diuron > 2,4-dichlorophenoxyacetic acid (2,4-D) (Figure 6B), with $q_{max}$ = 88, 36 and 5 mg g^−1^ at 25 °C and pH = 7.0, respectively. At this pH, the polymer surface is practically neutral (pH_PZC_ = 7.4) and atrazine and diuron are also neutral, so electrostatic interactions are not relevant. The decrease in material performance for 2,4-D removal with the increase of pH seems to indicate that electrostatic interactions play a role in its adsorption mechanism, since 2,4-D is negatively charged for pH > 2.8. Studies using salty simulated effluents proved that the ionic strength did not significantly affect the removal performance, again suggesting that adsorption is not limited by electrostatic interactions (Figure 6C). Besides this type of interaction, Fe@CF contains C–O bonds of esters, alcohols (in starch) and phenols (in tannin), which can lead to the formation of hydrogen bonds with the N- and carboxyl-containing herbicides, and Fe impregnation tends to increase the aromatization/hydrophobicity of the final structure, enabling hydrophobic and π–π physical interactions with herbicides. Adsorption processes followed the Langmuir model; however, isotherm profiles did not fully saturate for higher studied concentrations, suggesting also the occurrence of multilayers. Thermodynamic analysis revealed spontaneous (${∆G}^{0}$ = −30 to −27 kJ mol^−1^ (atrazine), −29 to −25 kJ mol^−1^ (diuron) and −22 to −21 kJ mol^−1^ (2,4-D)), and entropy-increasing (${∆S}^{0}$ = 119 J K^−1^ mol^−1^ (atrazine), 214 J K^−1^ mol^−1^ (diuron) and 4 J K^−1^ mol^−1^ (2,4-D)) processes, endothermic for atrazine (${∆H}^{0}$ = 8 kJ mol^−1^) and diuron (${∆H}^{0}$ = 39 kJ mol^−1^), and exothermic for 2,4-D (${∆H}^{0}$ = −20 kJ mol^−1^) (Table 3). The ${∆H}^{0}$ < 40 kJ mol^−1^ and the relatively fast adsorption ($t_{eq}$ = 90–240 min) again suggested the predominance of physisorption mechanism. Furthermore, density functional theory (DFT) computational calculations were performed and agreed with the experimental results, showing that the highest adsorption capacity of atrazine can be assigned to its higher chemical reactivity (owing to a small gap between HOMO and LUMO orbitals—Figure 6D–F) and neglected steric hindrance effects.

The versatility of starch-based materials was also efficiently extended to the removal of pharmaceuticals. Among the works reported in Table 3, the use of starch modified NiFe layered double hydroxide composite (S/NiFe-LDH, 1:1 ratio), prepared by coprecipitation, stands out for its enormous maximum removal capacity of the non-steroidal drug piroxicam ($q_{max}$ = 2840 mg g^−1^ at R_S-L_ = 0.5 g L^−1^ and natural pH = 8.3), possibly due to the rough and porous microstructure of S/NiFe-LDH, with S_BET_ = 6.7 m^2^ g^−1^ and $d_{P}$ = 3.4 nm, promising for an enhanced pollutant capture [100]. The adsorption of the drug followed the Freundlich (F) model (Figure 6G), showing a heterogeneous adsorption behavior. S/NiFe-LDH was also applied for the sorption of reactive blue 19 and reactive orange 16 dyes, showing $q_{max}$ of 149 and 182 mg g^−1^, respectively, estimated by the Langmuir model that best fitted the experimental data, pointing out a homogeneous monolayer adsorption, and that S/NiFe-LDH has both energetically homogeneous and heterogeneous adsorption sites. The kinetics for all analytes followed the PSO model (Figure 6H for piroxicam), suggesting a chemisorption mechanism, explained by the possibility of electrostatic interactions formation between the anionic piroxicam [225] or the SO_3_^−^ groups of the anionic dyes and the metal ions in S/NiFe-LDH. However, strong hydrogen bonds involving the O/N atoms of the adsorbates, π-cation interactions between polarized-π electron cloud of aromatic rings of pollutants and metal cations of the adsorbent, hydrophobic bonding involving alkyl chains of starch, Van der Waals interactions and some anion exchange of the nitrate anions of S/NiFe-LDH by negative dye molecules can also occur. Furthermore, S/NiFe-LDH was efficiently applied as a photocatalyst for the photodegradation of reactive red 120 dye, showing that the same material can be applied for two distinct but complementary applications in the pollutant remediation context, as in other recent works of starch-based materials reported in the literature (i.e., as adsorbents and catalysts [67,130,200], adsorbents and flocculants [49,136], or adsorbents and sensors [88]). It is also important to note that S/NiFe-LDH proved to be advantageous due to its excellent reusability and stability, showing only efficiency losses of 7%, 9%, 9%, and 14% after the fifth reuse cycle for the remediation of piroxicam, reactive blue 19, reactive orange 16 and reactive red 120 pollutants, respectively (Figure 6I).

## 5. Conclusions: Challenges and Prospects

The use of starch in environmental remediation, essentially by adsorption, is still underused. Although native starch is an abundant biopolymer in nature, it has some limitations, mainly with regard to its mechanical characteristics. Different strategies can be used to overcome these drawbacks. The most common are the functionalization of starch, the use of starch in composite materials or blends, or the use of starch to derivatize other polymeric materials, and thus being able to exploit the chemical characteristics of this polysaccharide. In general, the crosslinked, low-cost, biodegradable, and mainly mesoporous starch-based materials summarized have high potential for efficient removal of emerging pollutants, such as antibiotics or pesticides, as well as for applications in other areas such as imaging or drug delivery.

However, as a future prospect, we can think of starch as a polymer with multifunctional properties or greater selectivity. Among the possible approaches for the development of innovative starch-based materials, we can point out the possibility of developing materials with the capacity to remove pollutants and simultaneously act as probes and sensors. This could be achieved by synthesizing composite materials involving, e.g., carbon quantum dots—which makes the material luminescent — or by functionalizing the starch, taking advantage of the reactivity of its hydroxyl groups, with macrocyclic compounds, thereby increasing the capacity for supramolecular interaction and, consequently, the selectivity and the number of species that can be adsorbed. Alternatively, starch-based adsorbents could be formulated into membranes for filtration, used to prepare highly selective molecularly imprinted materials, or applied simultaneously in advanced oxidation processes. For example, Moreira et al. [138] envisaged the combined application in the future of an iron-functionalized rigid starch-based carbon foam for adsorption and photo-Fenton processes, to simultaneously remove and degrade pollutants.

It is also worth noting that the application of starch-based absorbents in large scale typically does not aim for high selectivity, but rather for an “universal adsorbent” that can remove the numerous pollutants present in effluents. This is a major challenge for the future. However, we believe that modifying starch to simultaneously introduce different functionalities into a final material could be the way forward to obtain more versatile and improved adsorbents, as revealed by some of the works reported in this review. The concerted incorporation of acidic and basic groups and aromatic units together with the hydroxyl groups of starch can allow the removal of a plethora of species (i.e., anionic, cationic and neutral simultaneously) through various types of interactions (i.e., coordination, electrostatic, Van der Waals, hydrogen, π-type and acid-base interactions, pore filling and ion exchange). In addition, the rational choice of modification agents will also be essential in order to achieve hierarchically structured materials preferably with the three types of porosity (micro-, meso- and macro-) and high surface area, as these properties greatly affect adsorption performance, since the size and volume of the cavities is important to allow the penetration and accommodation of the analytes into the material (and influencing the selectivity), and the intensity of this surface phenomenon is naturally proportional to the size of the interface (i.e., to the surface area).

## Figures and Tables

**Figure 1 polymers-17-00015-f001:**
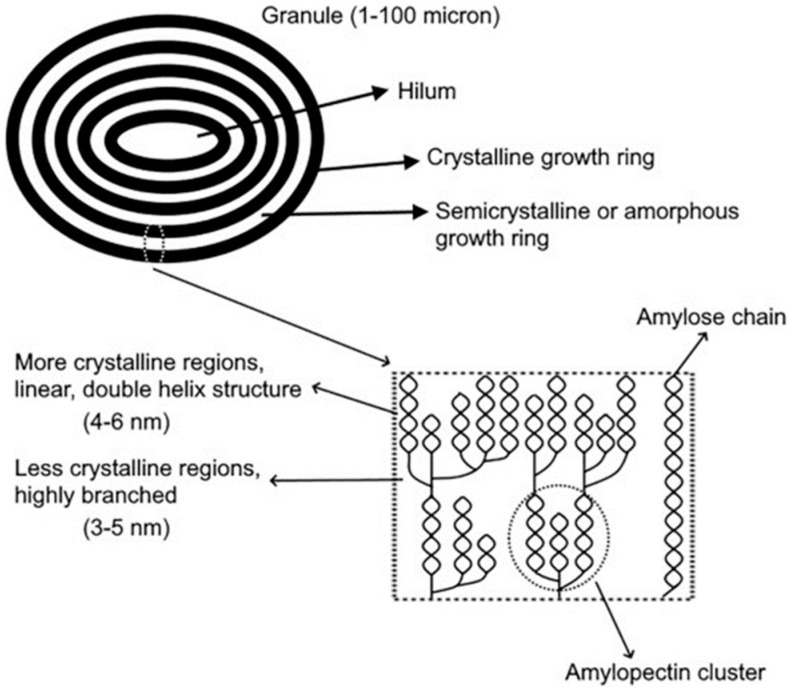
Organization of crystalline and amorphous regions in the granular structure of natural starches. Reprinted from [30] and reproduced with permission from Elsevier (Copyright © 2020, Elsevier).

**Figure 2 polymers-17-00015-f002:**
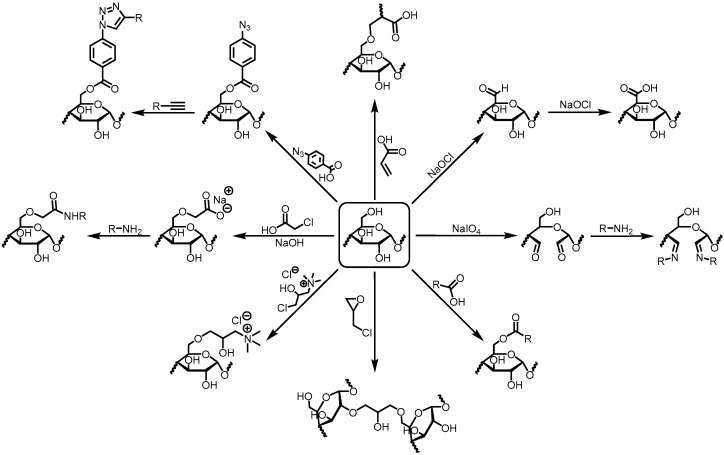
Examples of some chemical modifications performed on native starch.

**Figure 3 polymers-17-00015-f003:**
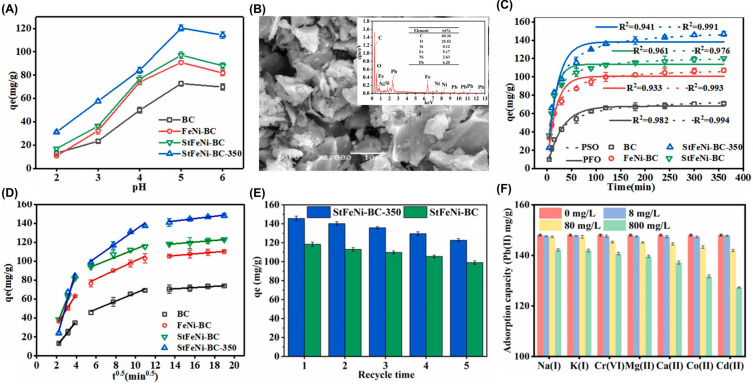
Adsorption of Pb(II) on StFeNi-BC-350: (**A**) Effect of pH, (**B**) SEM-EDX spectrum after Pb(II) adsorption; Fits of (**C**) PSO and PFO kinetic models, and (**D**) intraparticle diffusion model; (**E**) Reusability; and (**F**) Effect of interfering ions. Reprinted from [160] and reproduced with permission from Elsevier (Copyright © 2024, Elsevier).

**Figure 4 polymers-17-00015-f004:**
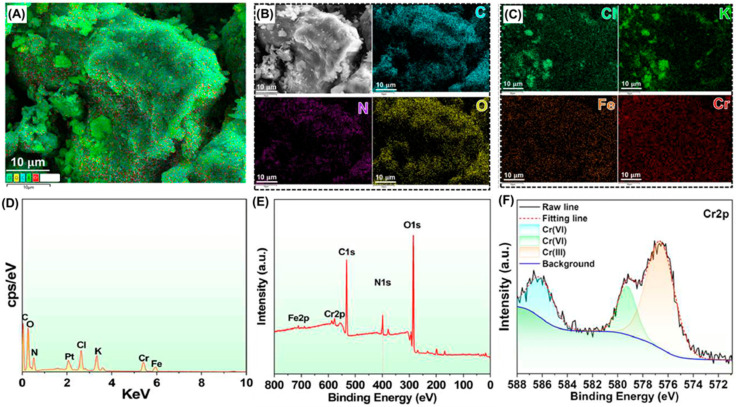
Adsorption of Cr(VI) on MAST: (**A**) SEM micrograph, (**B**) SEM-EDX mapping of C, O, N, and (**C**) Cl, K, Fe, Cr; (**D**) EDX, (**E**) XPS survey and (**F**) XPS (Cr 2p) spectra of post-adsorption MAST. Reprinted from [153] and reproduced with permission from Elsevier (Copyright © 2024, Elsevier).

**Figure 5 polymers-17-00015-f005:**
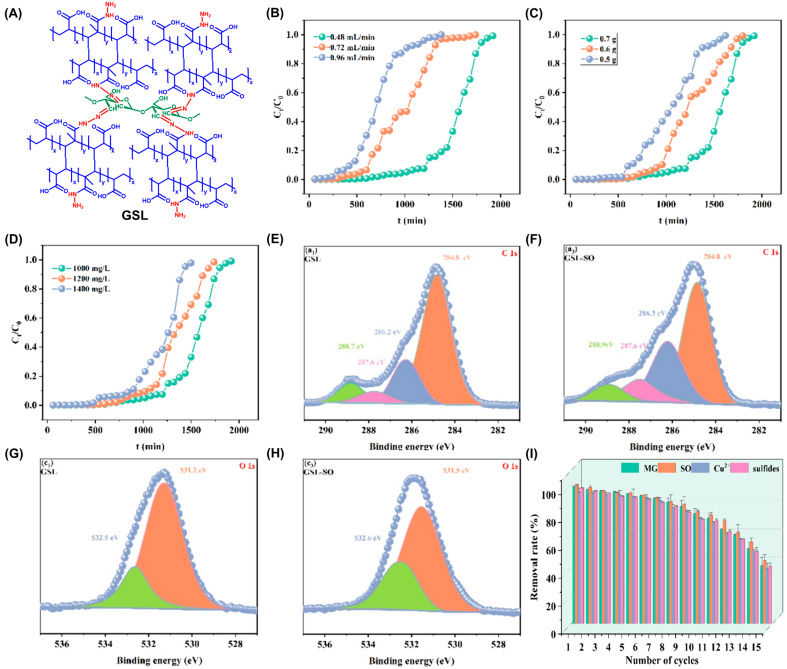
(**A**) Structure of GSL adsorbent; Effect of (**B**) flow rate, (**C**) adsorbent mass, and (**D**) inlet pollutant concentration on the breakthrough curve for SO continuous adsorption; (**E**–**H**) XPS spectra before (C 1s (**E**) and O 1s (**G**)) and after (C 1s (**F**) and O 1s (**H**)) SO adsorption; (**I**) reuse capacity of GSL. Reprinted from [54] and reproduced with permission from Elsevier (Copyright © 2024, Elsevier).

**Figure 6 polymers-17-00015-f006:**
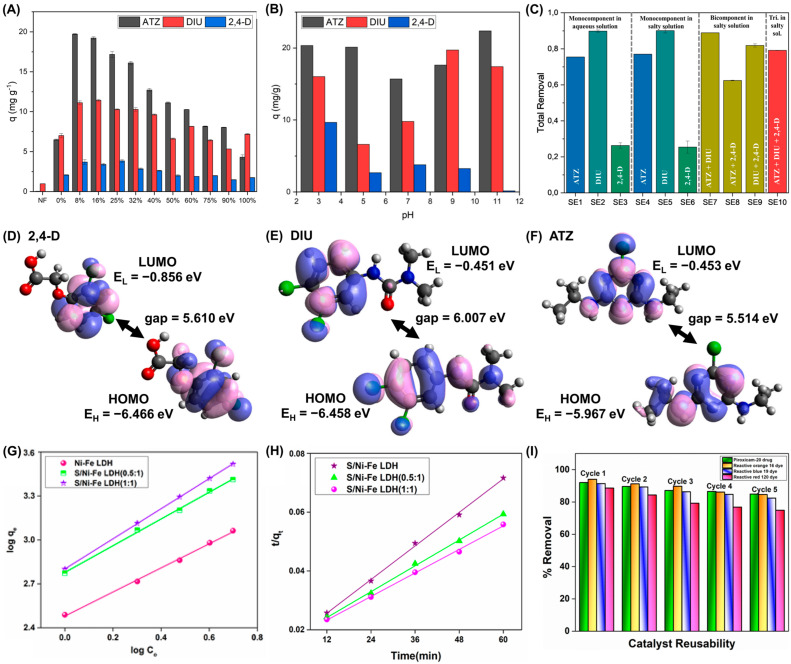
Adsorption using Fe@CF adsorbents: (**A**) Adsorption capacities of atrazine (ATZ), diuron (DIU), and 2,4-D herbicides using different Fe(III) content in the impregnation of rigid carbon foam; (**B**) Effect of pH on the adsorption capacities of each herbicide; (**C**) Adsorption performance for herbicide removal from simulated effluents; HOMO and LUMO orbitals of (**D**) 2,4-D (**E**) diuron and (**F**) atrazine. Reprinted from [138] and reproduced with permission from Elsevier (Copyright © 2023, Elsevier). Adsorption using S/NiFe-LDH adsorbent: Linear plots of (**G**) Freundlich isotherm and (**H**) PSO kinetic models for piroxicam adsorption; (**I**) reuse studies for the removal of piroxicam (green), reactive orange 16 (yellow), reactive blue 19 (blue), and reactive red 120 (pink). Reprinted from [100] and reproduced with permission from Springer Nature (Copyright © 2023, Springer Nature).

## Data Availability

Not applicable.

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
