# Peer review of "Recent Advances on Starch-Based Adsorbents for Heavy Metal and Emerging Pollutant Remediation"

_polymers, 2024, doi:10.3390/polym17010015_

Round 1
Reviewer 1 Report
Comments and Suggestions for Authors
The reviewed paper presents the latest results of work on the use of starch-based absorbents for the removal of heavy metals and pollutants. The state of research was presented in a reliable manner.
The disadvantage of the described adsorbents in mass use is their high selectivity. The summary of the research results lacked a discussion (longer than one sentence) about possible future modifications of starch leading to the production of a more universal adsorbent. Is this even possible? It would be desirable to present the Authors' opinion on this subject.
I have a few more minor comments:
-Line 214-215 The sentence is unclear.
-Table 1. I suggest explaining the symbols used by describing them in the table caption. The reader will not have to look for them in the main text.
-Thermodynamic parameters are given in Tables 1-3. However, I did not find their discussion in the text.
-line 464-465 „…described in Tables 1-3, they have thermal stability up to 150oC…” This sentence suggests that the temperature information is provided in the tables.
Author Response
The reviewed paper presents the latest results of work on the use of starch-based absorbents for the removal of heavy metals and pollutants. The state of research was presented in a reliable manner.
- The disadvantage of the described adsorbents in mass use is their high selectivity. The summary of the research results lacked a discussion (longer than one sentence) about possible future modifications of starch leading to the production of a more universal adsorbent. Is this even possible? It would be desirable to present the Authors' opinion on this subject.
We would like to thank the Reviewer for these pertinent questions. The conclusion of the manuscript has been improved to address the comment (Lines 875-897).
I have a few more minor comments:
- Line 214-215 The sentence is unclear.
In fact, the sentence was not clear. We have revised it and now reads: “The modification of the hydroxyl groups of starch involves overcoming physicochemical forces to make them accessible, since they are located within a layered microstructure”.
- Table 1. I suggest explaining the symbols used by describing them in the table caption. The reader will not have to look for them in the main text.
The ms has been improved accordingly.
- Thermodynamic parameters are given in Tables 1-3. However, I did not find their discussion in the text.
Thermodynamic parameters showed that most starch-based materials showed spontaneous, endothermic (with increased efficiency at higher temperatures) adsorption, and resulting in a random entropy increase. Some information is already described throughout the text about thermodynamics, such as in the following sentences of the first version of the manuscript:
- “The capture of Pb(II) by the composites proved to be a spontaneous, exothermic and entropy-decreasing process and to have good reusability.”
- “Thermodynamic parameters showed that all adsorption processes were spontaneous, endothermic (with increased efficiency at higher temperatures), and resulting in a random entropy increase.”
- “Thermodynamically, adsorption processes are spontaneous, endothermic and lead to an increase in entropy, suggesting a growth of the disorder at the solid/liquid interface.”
- “Thermodynamic analysis revealed spontaneous and entropy-increasing processes, endothermic for atrazine and diuron, and exothermic for 2,4-D (Table 3).”
However, we improved the discussion by adding more information about the thermodynamic parameters of reported works. Please, check the revised highlighted version of the manuscript.
- line 464-465 „…described in Tables 1-3, they have thermal stability up to 150oC…” This sentence suggests that the temperature information is provided in the tables.
The following information was added in the sentence “(results not reported in the Tables 1 – 3)” to clarify that the thermal degradation temperatures were not shown. In addition, the caption of Table 1 indicates that the temperatures in the Tables refer to adsorption conditions.
Reviewer 2 Report
Comments and Suggestions for Authors
Starch is one of the most abundant polysaccharides in nature and has a high potential for application in several fields, including effluent treatment, acting as an adsorbent. The author reviews the use of starch in the synthesis of different adsorbents: from nanoparticles to blends, and their performance in terms of amount of pollutant adsorbed and removal efficiency. This review is interesting but some corrections are needed:
1. In the abstract, the phrase “both the materials” in the sentence “A critical analysis of both the materials developed and the results obtained is 19 also carried out.” is confusing. Which both materials? It is hard to understand even going through the whole paragraph.
2. It is too wordy for the water pollution in the introduction. Much more attention should be put on the poly-69 saccharides.
3. English needs to be improved for better understanding.
4. The sentence in the fifth paragraph on the second page “Therefore, this paper reviews the recent progress (since 2023) in materials containing 77 neat or modified starch and other functional materials, regarding” is confusing. The above text tells that the author will focus on starch. Neat and modified starch still belong to starch things. However, what are “other functional materials”? What is their relationship with the starch?
5. there is a grammar mistake in the sentence in the fourth paragraph on the second page “and whose application 75 potential as adsorbent is still, in our view, underestimated. 76”.
6. The sentence in the second paragraph on the third page, “The 113 degree of crystallinity, determined by the amylose-to-amylopectin ratio and starch origin, 114 plays a critical role in influencing the functionality and processing behavior of starch 115 [23,24].“ should be put at the very beginning of this paragraph for better understanding.
7. A short conclusion or remarks should be given about the relation between “amylose/amylopectin” and “retrogradation” for better illustration.
8. In the part of “2.2. Properties of Starch”, the “crystallinity” should be illustrated as the first property, as the other two properties are both related with this one.
9. The first paragraph in the part “4. Application in contaminant removal” is confusing. The author should pay much more attention to illustrate applications of starch.
10. The author should denote the letter symbols in the Table 1 for easy reading.
Comments on the Quality of English LanguageThe English should be improved for better understanding.
Author Response
Comment 1. In the abstract, the phrase “both the materials” in the sentence “A critical analysis of both the materials developed and the results obtained is 19 also carried out.” is confusing. Which both materials? It is hard to understand even going through the whole paragraph.
Answer 1. We appreciate the Reviewer's comment. The sentence was modified.
Comment 2.It is too wordy for the water pollution in the introduction. Much more attention should be put on the polysaccharides.
Answer 2. Thank you for the comment. The Introduction was improved the introduction to contain more information about polysaccharides:
“Among the polymers that have attracted greater attention from researchers are polysaccharides, a relevant class of biological and biodegradable polymers obtained from renewable sources (i.e., plant, animal, microbe or algae) [19]; these include starch, gums (e.g., gum Arabic), cellulose (most abundant polysaccharide on Earth, based on D-glucose units linked by β(1→4) glycosidic bonds [16]), chitosan (the second most present polysaccharide in nature, being a linear cationic polymer consisting of D-glucosamine moieties and a few N-acetyl-D-glucosamine units [17]) and alginate (a linear polymer formed by β-D-manuronic and α-L-guluronic acid monomers [18]). Due to the nontoxicity, low cost, natural availability, biodegradability, biocompatibility and renewable character, these biopolymers are considered promising alternatives to traditional synthetic structures for many applications, including adsorption, catalysis and biomedical purposes [4]. In particular, this review focuses on a type of polysaccharide - starch - which is quite abundant in nature, since it is a primary storage polymer in plants [20,21], and has the capacity to be used as an adsorbent either on its own (with or without functionalization) [22], as a blend or composite [23], or even in nanoparticles [24,25], and whose application potential as adsorbent is still, in our view, underestimated.”. Additionally, a paragraph related with emerging pollutants (antibiotics) was deleted.
Comment 3. English needs to be improved for better understanding.
Answer 3. The English was proofread by a native speaker.
Comment 4. The sentence in the fifth paragraph on the second page “Therefore, this paper reviews the recent progress (since 2023) in materials containing 77 neat or modified starch and other functional materials, regarding” is confusing. The above text tells that the author will focus on starch. Neat and modified starch still belong to starch things. However, what are “other functional materials”? What is their relationship with the starch?
Answer 4. We appreciate the questions raised by the Reviewer. In fact, the reasons to use “other functional materials” are described in section 2.3 and in the first paragraph of section 3. Since starch presents some limitations, the introduction of other functional materials together with starch is a helpful strategy to obtain starch-based materials with improved properties and performances. To make this idea clearer, the following information “(used to overcome starch limitations)” has been added to the last sentence of the introduction.
Comment 5. there is a grammar mistake in the sentence in the fourth paragraph on the second page “and whose application 75 potential as adsorbent is still, in our view, underestimated. 76”.
Answer 5. The grammar mistake was corrected and the sentence reads now: “whose potential for application as an adsorbent is still, in our view, underestimated.”.
Comment 6. The sentence in the second paragraph on the third page, “The 113 degree of crystallinity, determined by the amylose-to-amylopectin ratio and starch origin, 114 plays a critical role in influencing the functionality and processing behavior of starch 115 [23,24].“ should be put at the very beginning of this paragraph for better understanding.
Answer 6. Following the Reviewer’s suggestion, the sentence was displaced.
Comment 7. A short conclusion or remarks should be given about the relation between “amylose/amylopectin” and “retrogradation” for better illustration.
Answer 7. A short conclusion stating “Consequently, starches with higher amylose content tend to retrograde faster and more extensively than those with higher amylopectin content, influencing the texture and shelf life of starch-based products” has been included in the Manuscript.
Comment 8. In the part of “2.2. Properties of Starch”, the “crystallinity” should be illustrated as the first property, as the other two properties are both related with this one.
Answer 8. In the revised version of the manuscript, the “Crystallinity” is illustrated as the first property.
Comment 9. The first paragraph in the part “4. Application in contaminant removal” is confusing. The author should pay much more attention to illustrate applications of starch.
Answer 9. We appreciate the Reviewer's comment. Taking the comment into account, the following information has been added to the first paragraph of part 4: “Although starch can be used efficiently in various applications (including food packaging and biodegradable films, drug and fertilizer delivery, emulsion, catalysis [178–182]), nowadays there is a huge opportunity to obtain promising starch-based structures for adsorption in order to mitigate environmental and water pollution, as will be described in the following subsections.”.
Comment 10. The author should denote the letter symbols in the Table 1 for easy reading.
Answer 10. The description of the symbols in the caption of Table 1 has been added.
Comment 11. Comments on the Quality of English Language. The English should be improved for better understanding.
Answer 11. The English was proofread by a native speaker.
Reviewer 3 Report
Comments and Suggestions for Authors
I am reporting the review results on the manuscript entitled: "Recent advances on starch-based adsorbents for heavy metal and emerging pollutant remediation". The manuscript presents a comprehensive assessment of current advances in starch-based adsorbents for heavy metal and emerging pollution remediation. The authors did an excellent job of gathering and assessing the relevant literature, emphasizing the potential of starch-based polymers as sustainable and effective adsorbents. Nevertheless, there are areas in which the manuscript could be further enhanced. This work can be accepted for publication in "Polymer journal" after a minor revision, which the comments have been raised up below:
1. The literature review is extensive and systematic. However, it may benefit from a more rigorous examination of the papers cited.
2. The authors should address the constraints and challenges of using starch-based adsorbents, including stability issues and probable functional group leaching.
3. The authors should elaborate on the mechanisms behind the improved adsorption capabilities of modified starch-based adsorbents.
4. A fuller description of the effect of surface functional groups and pore structure in adsorption would be beneficial.
5. The authors should investigate the effectiveness of several regeneration approaches, including thermal, chemical, and biological treatments.
6. The examination of the economic and environmental implications of starch-based adsorbents is brief.
7. Further research of the cost-effectiveness of these materials compared to traditional adsorbents would be beneficial.
8. The authors should present the possibility of combining starch-based adsorbents with other technologies like advanced oxidations and membrane filtration.
Comments on the Quality of English Language
English needs minor revision.
Author Response
Comment 1. The literature review is extensive and systematic. However, it may benefit from a more rigorous examination of the papers cited.
Response 1. We appreciate the Reviewer's comment. In fact, this literature review covers articles published since the beginning of 2023 on starch-based materials as adsorbents for environmental remediation, which is quite an extensive topic. Our aim with this review was to search for these different articles, select the most relevant ones, compile the synthesis, properties and the results of adsorption (isotherms, kinetics and thermodynamics) of the materials that led to the best performance among all those cited in the period (Tables 1 – 3). Finally, we described the key conclusions of some representative works in more detail to focus on different and relevant aspects related to the understanding of adsorption processes. In this sense, we consider that we conducted a rigorous examination of the papers and a careful selection of those that have been described in detail. Nevertheless, improvements have been made throughout the text to clarify some relevant points.
Comment 2. The authors should address the constraints and challenges of using starch-based adsorbents, including stability issues and probable functional group leaching.
Response 2. This question was addressed for the examples that have been described in more detail. Please, check the following sentences in the revised version of the manuscript:
- “As the reuse cycles increased, the adsorption performance gradually decreased. This might be due to the loss of adsorption sites of the adsorbent during regeneration.”
- “However, an increase in the number of adsorption cycles can lead to a decline in adsorption capacity, which could be attributed either to the saturation of active sites on the adsorbent's surface or partial damage to the adsorbent structure, resulting in a gradual reduction in MAST adsorption capacity. Thus, the choice of the conditions to carry out adsorption/desorption cycles must be fine-tuned to minimize structural changes, mass losses, and leaching/inactivation of functional groups, to achieve quantitative desorption efficiencies when adsorption is reversible, and to ensure that the material maintains its performance over many cycles (which will be important for minimizing secondary pollution caused by its disposal). Reaching all these conditions simultaneously is a huge challenge, since the optimization of the operation parameters is performed for each individual system and involves a compromise between the different variables.”
Comment 3. The authors should elaborate on the mechanisms behind the improved adsorption capabilities of modified starch-based adsorbents.
Response 3. In fact, the mechanism behind the improved adsorption capacities of the materials selected for more detailed description has been revealed using, as a basis, the characterization of those materials before and after the adsorption by FTIR, XPS, SEM, and EDX. In general, and as mentioned in Section 5. (“Conclusions: challenges and prospects”), the modification of starch allows the synthesis of materials with characteristics closer to our definition of “universal adsorbent”, since the modified starch-based adsorbents obtained are multifunctional and have improved morphological and surface properties for capturing various types of pollutants through different types of interactions. Also in Section 3.2, a sentence has been added that mentions the different aspects and analyses that must be considered in order to be able to predict the adsorption mechanism: “The assessment of the adsorbent before and after adsorption is a key point to understand the processes and mechanisms of the adsorption of pollutants from water using starch-containing bio-based adsorbents with improved performance, together with the effect of experimental parameters (pH, contact time, concentration of pollutants, temperature, interferents, and material regeneration) on adsorption, the analysis of the isotherm, kinetic and thermodynamic of adsorption, and the results of computational simulations, as will be described in Section 4.”
Comment 4. A fuller description of the effect of surface functional groups and pore structure in adsorption would be beneficial.
Response 4. In fact, we found limitations on this point. Even so, to address this issue, a more complete description has been provided throughout the manuscript and in the conclusion.
Comment 5. The authors should investigate the effectiveness of several regeneration approaches, including thermal, chemical, and biological treatments.
Response 5. We thank the Reviewer for this suggestion. This topic was added in section 4.2 of the manuscript. Please, check the following sentences in the revised version of the manuscript:
- “Thus, the choice of the conditions to carry out adsorption/desorption cycles must be fine-tuned to minimize structural changes, mass losses, and leaching/inactivation of functional groups, to achieve quantitative desorption efficiencies when adsorption is reversible, and to ensure that the material maintains its performance over many cycles (which will be important for minimizing secondary pollution caused by its disposal). Reaching all these conditions simultaneously is a huge challenge, since the optimization of the operation parameters is performed for each individual system and involves a compromise between the different variables.”
- “Between adsorption cycles, the previous adsorbent (GSL) was chemically regenerated using a 1 mol dm-3 HCl solution [54]. In general, chemical desorption using different eluents (acid, bases, chelating agents, salts, organic solvents) is the preferred strategy for the regeneration of starch-based materials and numerous types of novel adsorbents synthesized in recent years [193,219,220]. However, other approaches can also be employed, albeit less frequently, such as thermal regeneration (typically for conventional adsorbents, e.g., activated carbon or zeolites, since starch-containing materials and organic-based adsorbents degrade at aggressive temperature conditions [219,221]) or bioregeneration/biodegradation (i.e., by microorganism activity and enzymatic reactions [222,223]). It is also important to note that economic and environmental issues have to be taken into account when searching for an appropriate desorption method, since regeneration at high temperatures involves high energy costs; while elution can involve unsafe and unsustainable substances, and require additional steps to regenerate (e.g., neutralization) the material before a new adsorption cycle. In this sense, an advantageous approach to explore is the eluent reuse in pollutant desorption and concentration [224].”
Comment 6. The examination of the economic and environmental implications of starch-based adsorbents is brief.
Response 6. The examination of the environmental and economic issues of starch-based adsorbents was complemented. Please, check the following sentences in the resived version of the manuscript:
- “In this sense, the percentage of modification and its relationship in terms of benefit to the final performance of the material must be studied rigorously to avoid making the material too expensive.”
- “To get further insights about economic viability, some authors have estimated the cost of material production. For instance, Zhang et al. [161] estimated a cost of 5.91 USD/kg to produce magnetic functionalized carbon microsphere (MF-CMS) and that, per kg of adsorbent, 8 kg of waste rice are consumed and 0.55 tons of tetracyclines (TCs) can be treated in wastewaters. Additionally, desorption of TCs was achieved using 0.1 M NaOH and MF-CMS was regenerated using methanol and reused over 5 cycles, showing a removal rate decrease of only 20% and excellent stability and environmental safety. So, the cost of MF-CMS was comparable to commercial materials (such as activated carbon), however, the traditional adsorbents, besides the high cost, are difficult to regenerate and show a significant reduction in adsorptive capabilities after continuous regeneration cycles [161,172].”
- “…to ensure that the material maintains its performance over many cycles (which will be important for minimizing secondary pollution caused by its disposal).”
- “It is also important to note that economic and environmental issues have to be taken into account when searching for an appropriate desorption method, since regeneration at high temperatures involves high energy costs; while elution can involve unsafe and unsustainable substances, and require additional steps to regenerate (e.g., neutralization) the material before a new adsorption cycle. In this sense, an advantageous approach to explore is the eluent reuse in pollutant desorption and concentration [224].”
Comment 7. Further research of the cost-effectiveness of these materials compared to traditional adsorbents would be beneficial.
Response 7. The cost-effectiveness description has been improved (please, check Section 3.2).
Comment 8. The authors should present the possibility of combining starch-based adsorbents with other technologies like advanced oxidations and membrane filtration.
Response 8. This is a pertinent comment. The conclusion of the manuscript has been improved to contain a reference to that. For example, the following sentence has been added: “Alternatively, starch-based adsorbents could be formulated into membranes for filtration, used to prepare highly selective molecularly imprinted materials or applied simultaneously in advanced oxidation processes. For example, Moreira et al. [138] envisaged the combined application in the future of an iron-functionalized rigid starch-based carbon foam for adsorption and photo-Fenton process, to simultaneously remove and degrade pollutants.”
Comment 9. Comments on the Quality of English Language. English needs minor revision.
Response 9. The English was revised and improved throughout the manuscript.
Round 2
Reviewer 2 Report
Comments and Suggestions for Authors
This manuscript has been much improved and can be accepted now.